| Evolution | Research Article

# Snowball: a novel gene family required for developmental patterning of fruiting bodies of mushroom-forming fungi (Agaricomycetes)

Csenge Földi,[1,2] Zsolt Merényi,[1] Bálint Balázs,[1] Árpád Csernetics,[1] Nikolett Miklovics,[1] Hongli Wu,[1] Botond Hegedüs,[1] Máté Virágh,[1] Zhihao Hou,[1] Xiao-Bin Liu,[1] László Galgóczy,[1,3] László G. Nagy[1]

**ABSTRACT** The morphogenesis of sexual fruiting bodies of fungi is a complex process determined by a genetically encoded program. Fruiting bodies reached the highest complexity levels in the Agaricomycetes; yet, the underlying genetics is currently poorly known. In this work, we functionally characterized a highly conserved gene termed *snb1*, whose expression level increases rapidly during fruiting body initiation. According to phylogenetic analyses, orthologs of *snb1* are present in almost all agaricomycetes and may represent a novel conserved gene family that plays a substantial role in fruiting body development. We disrupted *snb1* using CRISPR/Cas9 in the agaricomycete model organism *Coprinopsis cinerea*. *snb1* deletion mutants formed unique, snowball-shaped, rudimentary fruiting bodies that could not differentiate caps, stipes, and lamellae. We took advantage of this phenotype to study fruiting body differentiation using RNA-Seq analyses. This revealed differentially regulated genes and gene families that, based on wild-type RNA-Seq data, were upregulated early during development and showed tissue-specific expression, suggesting a potential role in differentiation. Taken together, the novel gene family of *snb1* and the differentially expressed genes in the *snb1* mutants provide valuable insights into the complex mechanisms underlying developmental patterning in the Agaricomycetes.

**IMPORTANCE** Fruiting bodies of mushroom-forming fungi (Agaricomycetes) are complex multicellular structures, with a spatially and temporally integrated developmental program that is, however, currently poorly known. In this study, we present a novel, conserved gene family, Snowball (snb), termed after the unique, differentiation-less fruiting body morphology of snb1 knockout strains in the model mushroom Coprinopsis cinerea. snb is a gene of unknown function that is highly conserved among agaricomycetes and encodes a protein of unknown function. A comparative transcriptomic analysis of the early developmental stages of differentiated wild-type and non-differentiated mutant fruiting bodies revealed conserved differentially expressed genes which may be related to tissue differentiation and developmental patterning fruiting body development.

**KEYWORDS** mushroom development, unannotated genes, Basidiomycota, tissue differentiation, *Coprinopsis cinerea*, fruiting body

The development of sexual fruiting bodies in the Agaricomycetes (mushroom-forming fungi) is one of the most complex ontogenetic processes in the fungal kingdom (1–4). Fruiting bodies are complex multicellular structures that facilitate the dispersal of sexual spores by sophisticated and often cryptic mechanisms, such as lifting the hymenophore (spore-producing tissue) above ground (2, 5), utilizing the power of wind

Address correspondence to László G. Nagy, lnagy@fungenomelab.com.

The authors declare no conflict of interest.

See the funding table on p. 21.

or forcible spore discharge, among others. The emergence of three-dimensional fruiting bodies on the vegetative mycelium, which has fractal-like dimensions, represents a transition in complexity level and is a developmental process regulated by a genetically encoded program. At the initiation of fruiting body development, hyphal aggregates, hyphal knots, appear on the vegetative mycelium. These structures develop into stage 1 primordia, in which developmental patterning first becomes visible with the emergence of tissues representing the cap, stipe, and veil (7). Deciphering the molecular networks behind fruiting body development is a century-old challenge in mycology. Several aspects of fruiting body development are already known, from classic histology (6), to forward and reverse genetics or other approaches (7). These include the importance of structural (e.g., hydrophobins (8), cell wall biosynthetic and modifying enzymes (9–11) and regulatory genes (e.g., transcription factors (12–16), chromatin remodelers (17, 18), kinases (19), or signal transduction genes (20, 21). However, the genetic knowledge accumulated over the years covers only a small fraction of what conceivably constitutes the developmental program and does not currently allow general principles that are applicable across species to shine through individual studies (22).

The recent proliferation of -omics studies in the Agaricomycetes has uncovered a large number of developmentally regulated genes, defined as ones that show considerable expression dynamics during development or between tissue types (23–27). However, there is currently a large gap between -omics studies and genetic analysis of developmental genes. While the former can identify hundreds to thousands of developmentally regulated genes, functional analysis of these is lagging, with orders of magnitudes fewer genes with a known function, in any species (28). Uncovering the precise function of these genes could fill the gaps in our understanding of the complex molecular networks underlying fruiting body development.

A particularly interesting group of developmentally regulated genes encode proteins without known conserved domains (e.g., Pfam, InterPro) or with domains of unknown function (DUF). The functional annotation of these genes is difficult by commonly used homology-based methods, and represent the "dark proteome" encoded by the genomes of various organisms. Such genes might constitute new families or even novel folds with hitherto unknown functionalities, as demonstrated recently across the whole protein universe (29) or in the context of fungal examples (30). Unannotated genes are widespread in fungal genomes and can represent truly novel gene families but also spurious open reading frames that stem from gene annotation artifacts (31). Thus, although genes without known functions can conceivably encode new functionalities, unlocking their function is a complex problem.

The latest developmental -omics studies identified hundreds of unannotated and poorly known genes in fruiting body transcriptomes, many of which displayed considerable sequence conservation and even showed conserved expression patterns. A recent analysis of twelve mushroom-forming species found that a large proportion of conserved developmentally regulated genes is, actually, unannotated or encode proteins with DUFs and that the genome of *Coprinopsis cinerea* alone contains 158 such families (28). On the other hand, less than a handful of them have been functionally characterized in mushroom-forming fungi. Notable examples related to fruiting body development include *PriA*, which is strongly induced in primordia *Lentinula edodes* but its exact functions are unknown (32), or *Spc14* and *Spc33* which are involved in the formation of the septal pore cap in *Schizophyllum commune* (33). Despite these examples, the vast majority of these genes remain in the dark and thus the roles they fulfill during development are also left uncharted.

In this study, we set out to functionally characterize a gene of unknown function in *C. cinerea,* a widely used model system in fungal developmental biology (7). We identified *C. cinerea 493979* which exhibits high levels of conservation in the Agaricomycetes and based on its expression patterns across species, may play a role in fruiting body development. We show that its deletion results in severely stunted, snowball-like fruiting bodies. Based on the phenotype, we term it *snb1* (*'snowball-1'*) and find that the

fruiting defect stems from the failure of tissue differentiation in the primordia. We take advantage of this undifferentiated phenotype to gain insights into genes related to the emergence of tissues within fruiting bodies by RNA-Seq analyses.

## RESULTS

A recent study identified 921 conserved developmentally expressed gene families in 12 agaricomycete species, of which 158 represent orthogroups with unknown function (28). Among these, the *C. cinerea snb1* gene (*C. cinerea* AmutBmut1 strain protein ID: *493979*, *C. cinerea* Okayama-7 protein ID: CC1G_01874) stands out because its orthologs showed developmentally dynamic expression in *C. cinerea* and the other 11 species. In *C. cinerea,* the expression of *snb1* is negligible in vegetative mycelia but increased 144- and 240-fold in the initial steps of fruiting body formation, secondary hyphal knots, and stage 1 primordia, respectively (26).

### Targeted disruption of *snb1*

To delete the *C. cinerea snb1* gene, we applied the CRISPR/Cas9 system, comprising a ribonucleoprotein (RNP) complex with a single guide RNA (sgRNA) and the Cas9 protein together with a DNA repair template (34). To perform this, we utilized the *C. cinerea* AmutBmut1 strain, which is a *p*-aminobenzoic acid (PABA) auxotrophic homokaryotic strain, which we denote here as the wild type (wt). We obtained three independent mutants, in which PCR analysis using primers upstream of the locus and in the *pab1* gene verified the integration of the repair template into the Cas9 cleavage site (Fig. S1).

### The lack of *C. cinerea snb1* results in spherical, poorly differentiated, snow-ball-like fruiting bodies

We measured the growth rate of vegetative mycelium on rich media (YMG and complete Fries) and under starvation conditions [Fries medium without carbon (-C) or nitrogen (-N) sources] (35). The growth rate of the wt and deletion (Δ*snb1*) strains did not differ significantly under these conditions (two-way ANOVA, $P = 0.460$). There was no significant difference between the wt and Δ*snb1* strains in either fresh or dry mass after 6 days of growth on YMG medium at 28°C (one-way ANOVA, $P = 0.993$ and $P = 0.353$, respectively, results not shown). All three independent Δ*snb1* strains behaved the same way.

The fruiting bodies of the Δ*snb1* strains differed remarkably from those of the wt. Under a synchronized fruiting protocol, in which a 2-h light irradiation triggers fruiting (24), both wt and Δ*snb1* strains produced well-defined rings of secondary hyphal knots, approximately 24 h pLI (Fig. S2). However, while wt hyphal knots developed into stage 1 and 2 primordia with well-discernible cap, stipe, gill, and universal veil, all independent Δ*snb1* strains produced white, spherical, snowball-like structures with apparently no or little differentiation (Fig. 1). Based on this morphology, we termed *C. cinerea* ID: *493979* gene *snb1*, referring to the snowball-like appearance of mutant fruiting bodies. As development progressed, rudimentary, apically formed cell-dense structures became visible on the fourth day pLI in cross-sections of fruiting bodies (Fig. 1). The fruiting bodies were unable to form cap, stipe, and gills; hence, they were not able to sporulate. Nevertheless, the fruiting bodies remained spherical with apical structures resembling the rudiments of the universal veil present. Due to the absence of a cap, the partial veil was missing. Consequently, Δ*snb1* strains were also not able to produce mature fruiting bodies and spores (Fig. 1).

The reintroduction of the wt *snb1* gene into the Δ*snb1* strains rescued the mutant phenotype. The integration of the wt *snb1* gene was validated by PCR (Fig. S3A). The complemented Δ*snb1* strains were able to produce fruiting bodies that were identical to those of the wt strain (Fig. S3B).

Without light induction, wt *C. cinerea* primordia develop into dark stipes, which results from the elongation of the basal part (nodulus), while the stipe and cap remain

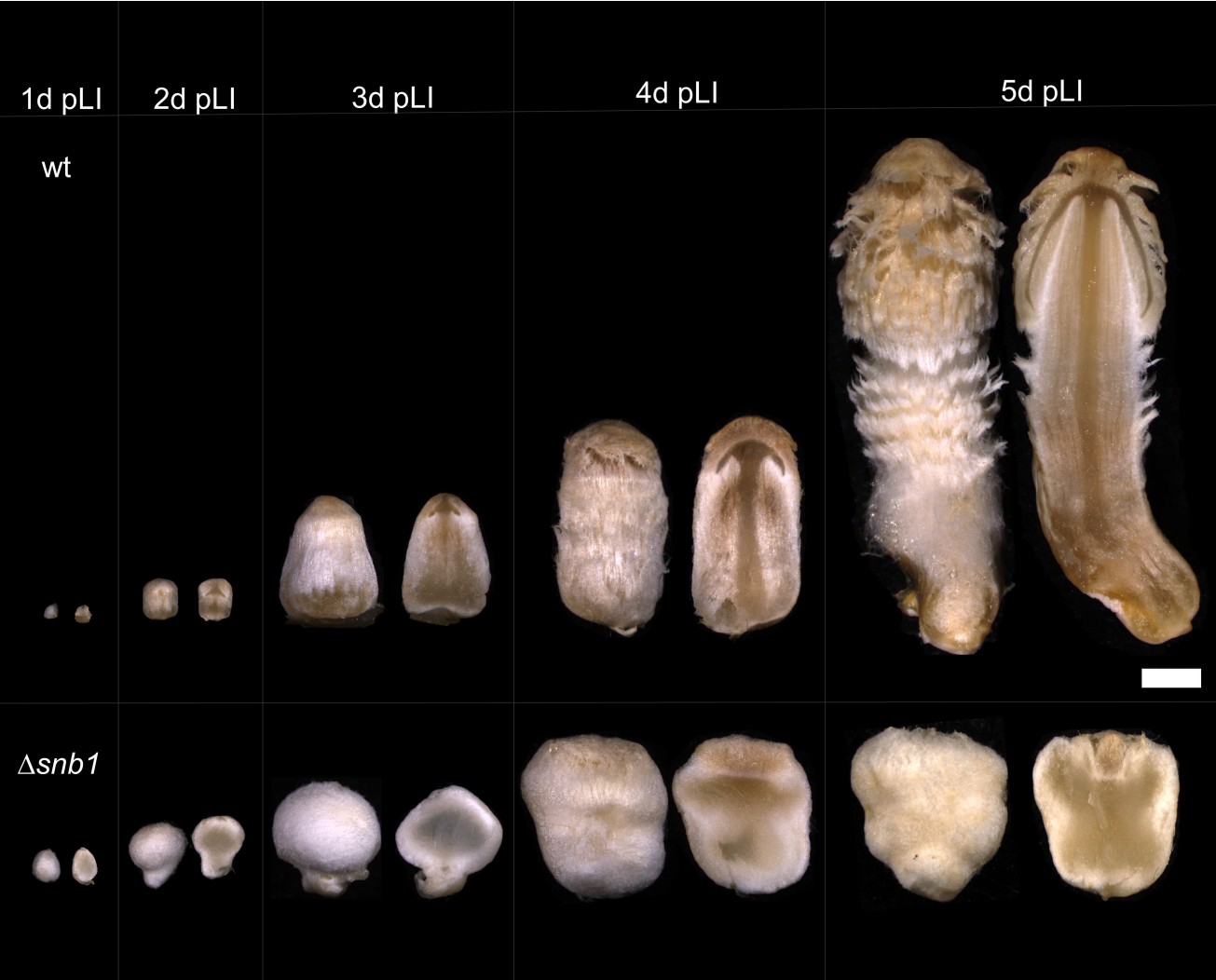

**FIG 1** Cross-sections of developing fruiting bodies of wt and Δ*snb1 Coprinopsis cinerea* strains during the first 5 days post-light induction (pLI). Strains were grown on YMG medium with halved glucose content at 28°C. Scale bar = 1 mm.

rudimentary (Fig. 2) (36). The Δ*snb1* strains were not able to produce dark stipes under constant dark conditions, rather, they produced spherical fruiting structures scattered on the vegetative mycelia. These observations suggest that the mechanisms responsible for nodulus elongation in the wt strain are inactive in the Δ*snb1* strains.

## Δ*snb1* strains show higher tolerance to the fruiting inhibitor LiCl

LiCl suppresses fruiting body formation presumably through the inhibition of glycogen synthase kinase-3 (GSK-3) (37). To test whether Δ*snb1* is sensitive to LiCl, we investigated the effect of LiCl on wt and Δ*snb1* fruiting body development (Fig. 3). The untreated wt developed mature fruiting bodies while fruiting body development stopped at stage 3 primordia in the presence of 0.5 g/l LiCl. This was a lower LiCl inhibitory concentration than expected based on the literature, yet the effect was similar to that previously reported (37). At higher LiCl concentrations (1–2 g/L), the growth of wt strain was arrested at the hyphal knot stage and did not continue even after two weeks of incubation. Therefore, we used a more resolved LiCl concentration gradient, up to 2 g/L. By contrast, fruiting body development was not inhibited in the Δ*snb1* strain up to 1 g/L LiCl. In these, an elongation of the basal part of the fruiting body was observed. At higher LiCl concentrations (1.5 and 2 g/L), Δ*snb1* primordia were also arrested approximately at

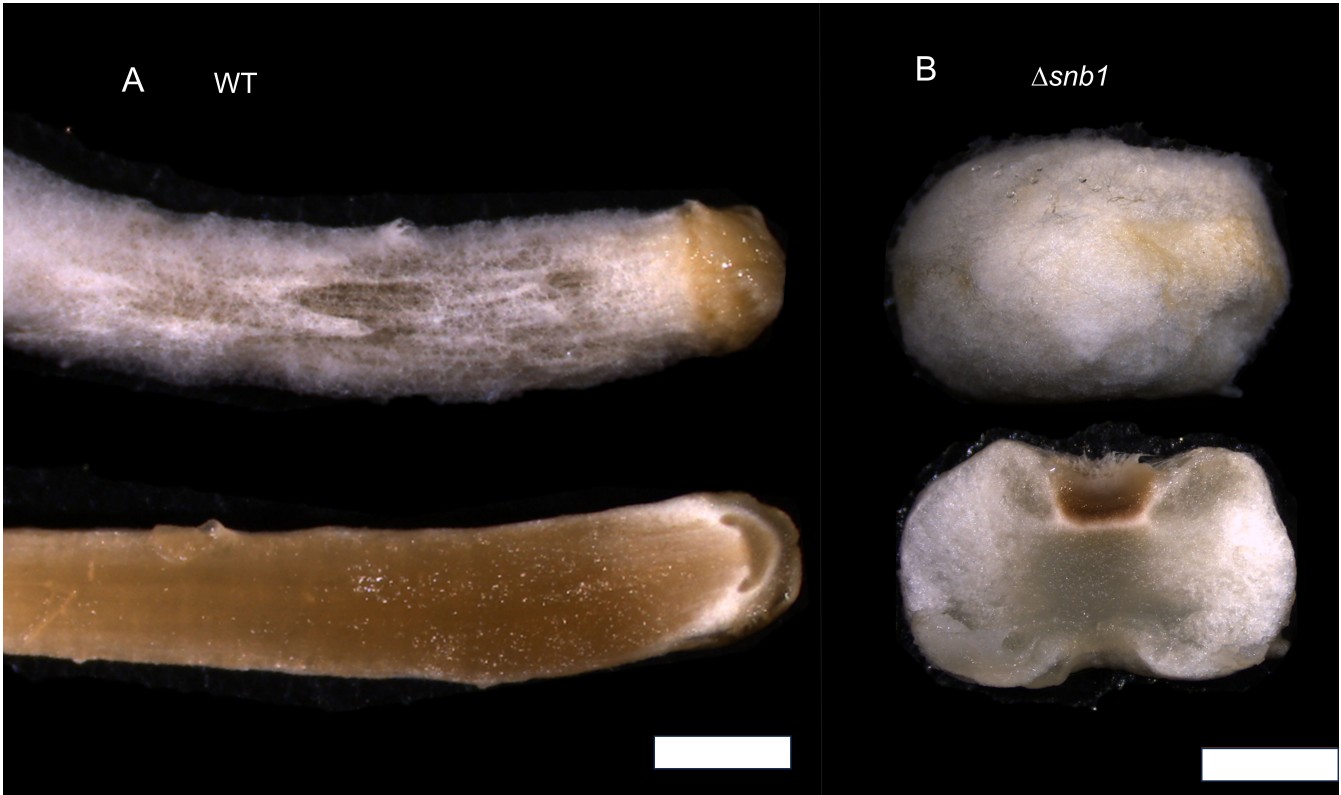

**FIG 2** Fruiting bodies of wt and Δ*snb1* *Coprinopsis cinerea* strains after 2 weeks of growth in the dark. (A) The wt fruiting bodies elongated in darkness (dark stipes tilted by 90°). (B) Δ*snb1* fruiting bodies expanded laterally and remained spherical, and were unable to produce dark stipes. Scale bar = 2 mm.

stage 1 primordium. Based on these observations, we conclude that the Δ*snb1* strain is less sensitive to the repressive effect of LiCl on fruiting body development.

## Domain composition and predicted structure of the SNB1 hypothetical protein

An InterPro search revealed that the predicted SNB1 comprises two disordered loop regions and a domain of unknown function (DUF6533; IPR045340). The latter is a basidiomycete-specific short, conserved region found in fungal integral membrane proteins [(38); version: InterPro 96.0]. WoLFPSORT predicted that SNB1 is localized in the plasma membrane. According to DeepTMHMM, it contains seven transmembrane helices with an extracellular N-terminus and an intracellular C-terminus (Fig. 4A and B). The tertiary structure of SNB1 was predicted by AlphaFold, which indicated the arrangement of the seven transmembrane domains and suggested a long, albeit poorly resolved intracellular loop region.

### Phylogenetic analysis of SNB1 orthologs

We used a published data set (22) to identify homologs of SNB1 across 109 fungal species. Phylogenetic analyses revealed that the gene family of SNB1 is absent in the Ascomycota, whereas it is conserved in the Basidiomycota (Fig. 4C).

The most distant homologs of SNB1 were identified in the Pucciniomycotina and Ustilaginomycotina; however, no homologs were found in the Tremellomycetes and Dacrymycetes. Most species of the Agaricomycetes contain 1–3 copies, while members of the Agaricales have slightly higher copy numbers. For example, Psathyrellaceae (where *C. cinerea* belongs) species have 4–5 homologs. A gene tree-species tree reconciliation analysis and mapping of ancestral gene copy numbers suggest that ancestral copy numbers have been relatively stable during agaricomycete evolution. We did not identify

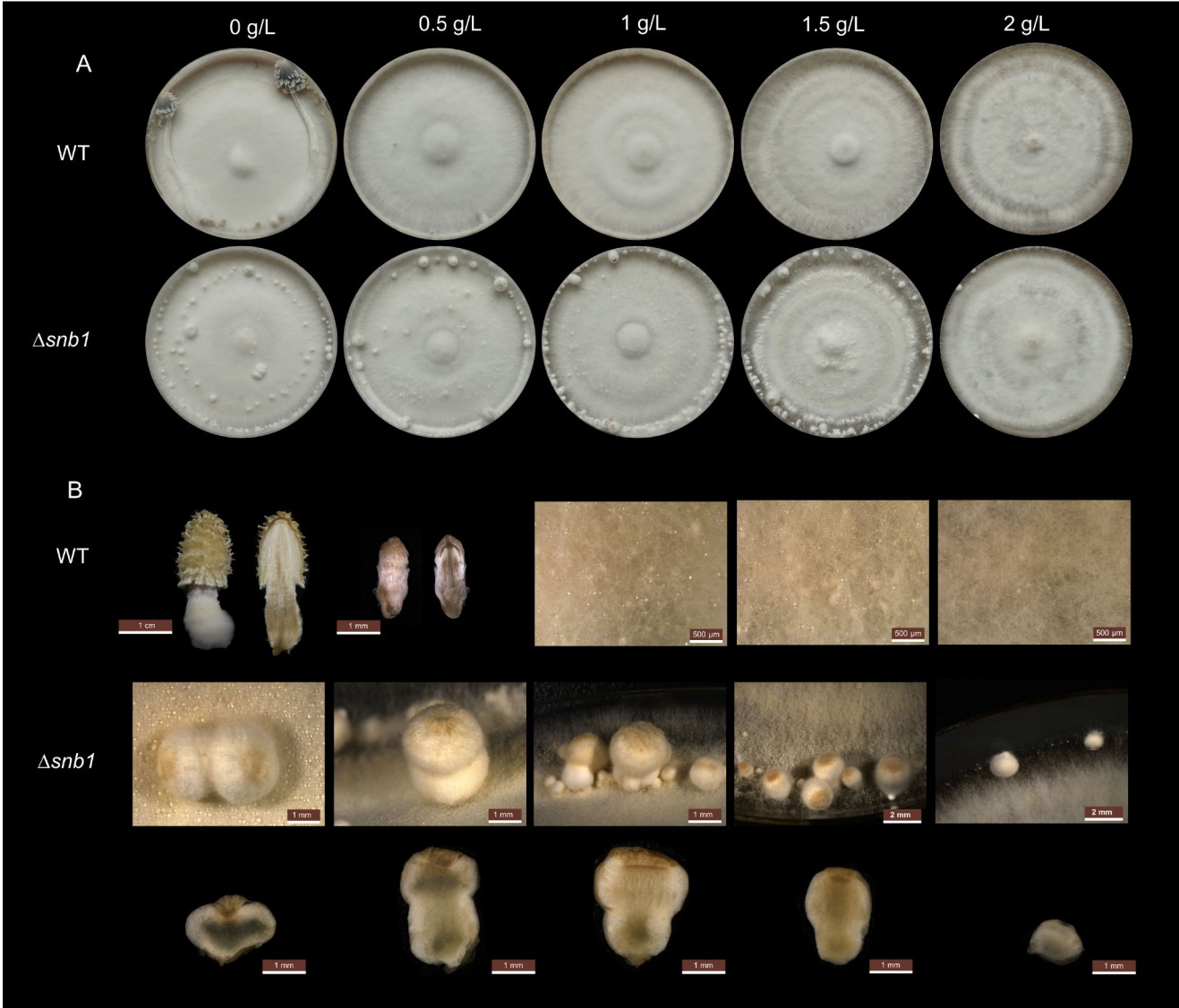

**FIG 3** The effect of LiCl on *Coprinopsis cinerea* fruiting body development. Pictures represent the final developmental stages after 2 weeks of incubation on a YMG medium with halved glucose content and increasing concentrations of LiCl. (A) Top view of wt and Δ*snb1* colonies. The vegetative mycelia of the strains were grown under alternating dark and light cycles, leading to the formation of ring-like structures on the mycelia. (B) Whole and sectioned fruiting bodies of wt and Δ*snb1* strain.

major expansions or contractions of the gene family in any larger taxon, except the loss of the family in Tremello- and Dacrymycetes. The genome of *C. cinerea* encodes four members of the *snb1* family (*snb1, 292504, 407873,* and *500548*). Among these only *snb1* was upregulated at the early stages of the fruiting body development, which suggests that its paralogs have other functions.

Among the homologs of SNB1, the orthogroup encompassing it originated in early Agaricomycetes and is found in extant taxa belonging to the Auriculariales, Tulasnellales, and more derived orders (Fig. S4A). Based on data published previously (22, 26, 39–43), we verified that orthologs of SNB1 in other species showed a similar expression pattern during fruiting body development (Fig. S4B), suggesting that SNB1 orthologs are universally dynamically expressed during the development of agaricomycete fruiting bodies. This observation implies that members of this orthogroup probably play a pivotal role in this process, ultimately forming a new gene family related to fruiting body development.

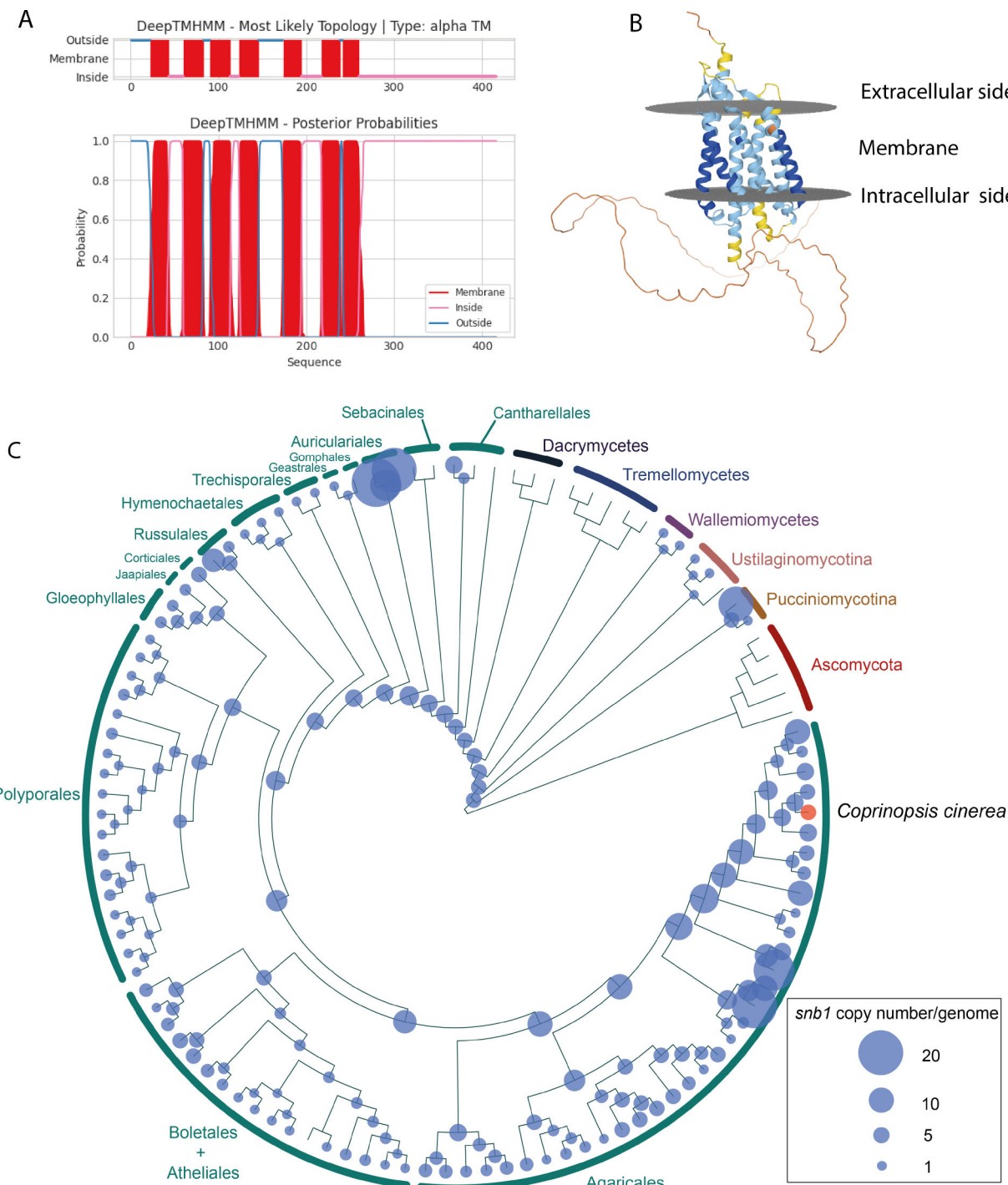

**FIG 4** Structural and phylogenetic analysis of SNB1. Based on predictions made by (A) DeepTMHMM and (B) AlphaFold (https://alphafold.ebi.ac.uk/entry/A8N2R0), SNB1 possesses seven transmembrane domains. The N-terminus of SNB1 is oriented toward the extracellular environment, while the C-terminus faces the intracellular space. (C) Maximum likelihood phylogenetic tree showing copy numbers of *snb1* in 109 species. Ascomycota was used as an outgroup. *Coprinopsis cinerea AmutBmut1* is highlighted with red, which has four *snb1* paralogs. For detailed data, see Fig. S4A.

## RNA-Seq analysis reveals transcriptomic changes in Δ*snb1* mutants during fruiting body initiation

Based on previous transcriptomic data sets, *snb1* expression is almost undetectable in vegetative mycelia but strongly increases during fruiting body initiation (26), which, in

conjunction with the observed phenotype, suggests that *snb1* may play an essential role in early events of fruiting body development. Our phenotyping data indicated that the Δ*snb1* strain is defective in the differentiation of cap and stipe tissues. We took advantage of this fact to get insights into the differentiation program of *C. cinerea* fruiting bodies by RNA-Seq analysis. To investigate the genes that are differentially expressed in the absence of SNB1 and thus might be associated with the lack of proper fruiting body differentiation, we obtained gene expression profiles during fruiting body initiation in the wt and the Δ*snb1* strain. Because the expression of *snb1* was negligible in the vegetative mycelium and there were no changes in the growth rate, we did not expect differentially expressed genes (DEGs) in that stage. Therefore, we examined the earliest, manually collectable developmental stages, secondary hyphal knots (wt: d =~ 200 µm, Δ*snb1*: d =~ 550 µm, 24 h pLI), which are undifferentiated in both strains, and stage 1 primordia (wt: d =~ 500 µm, Δ*snb1*: d =~ 750–900 µm, 48 h pLI), in three biological replicates (Fig. S2). Although secondary hyphal knots of the Δ*snb1* strain were slightly larger than those of wt, both were undifferentiated and had similar internal organization. By contrast, stage 1 primordia of the wt had clearly visible cap and stipe tissues, while those of Δ*snb1* remained undifferentiated (Fig. S2).

RNA-Seq analysis revealed a total of 1,299 DEGs that showed significant ($P ≦ 0.05$, fold-change (FC) >2) expression alteration in the Δ*snb1* strain compared to wt 24 h and 48 h pLI. At 24 h pLI, in the secondary hyphal knots, we found 177 downregulated and 218 upregulated DEGs in the Δ*snb1* strain. At 48 h pLI which corresponds to stage 1 primordium, we identified 516 downregulated and 650 upregulated DEGs (Table S2). In total, 574 DEGs and 727 DEGs were down- and upregulated significantly in at least one sample, respectively. A total of 267 genes were significantly differentially expressed ($P ≦ 0.05$) at both time points, among these 119 were downregulated and 141 were upregulated at both time points, and two were differently regulated at the two time points. The higher number of DEGs at 48 h pLI is consistent with the greater divergence of the morphologies of wt and Δ*snb1* strains as development progresses.

We functionally characterized DEGs using Gene Ontology (GO) enrichment analysis. This showed a significant ($P$ 0.05) overrepresentation of 93 GO terms (Fig. S5; Table S3), which were mostly generic terms that are hard to interpret in the context of fruiting body development. Therefore, we chose to characterize DEGs using comparisons to previously published data for *C. cinerea* and manual classification of developmentally regulated genes.

Krizsán et al. (2019) categorized genes as "FB-init" if their expression increased at least fourfold in hyphal knots and stage 1 primordia, relative to vegetative mycelium. We found that 236 downregulated DEGs (41.11%) belonged to the "FB-init" category, while only 31 upregulated DEGs (4.26%) could be classified into this group (Fig. 5A). This suggests that genes downregulated in Δ*snb1* are transcriptionally relevant to early developmental events, during which the main tissue types emerge.

Furthermore, we reevaluated the data set of Krizsán et al. (2019) to identify genes exhibiting tissue-enriched expression. For this, we used the young fruiting body stage, in which data are available separately for the cap, gills, and stipes. In total, we found 2,554 tissue-enriched genes. Of the Δ*snb1* DEGs, 582 were tissue enriched, 333 (58.01%) showed downregulation, and 250 (34.39%) upregulation in the Δ*snb1* strain. One DEG overlapped, this was significantly upregulated at 24 h pLI and downregulated at 48 h pLI. The high ratio of tissue-specific downregulated DEGs underpins our speculation that examining DEGs in the Δ*snb1* strain can reveal genes associated with tissue differentiation.

Upregulated genes could result either from processes becoming activated only in the mutant, or from differences in tissue mass ratios (i.e., transcript being relatively more dominant because the mutant is composed of a single tissue). With the latter being an RNA-Seq artifact, we think upregulated DEGs are less interesting from the perspective of differentiation. Therefore, in subsequent analyses, we pay more attention to downregulated DEGs.

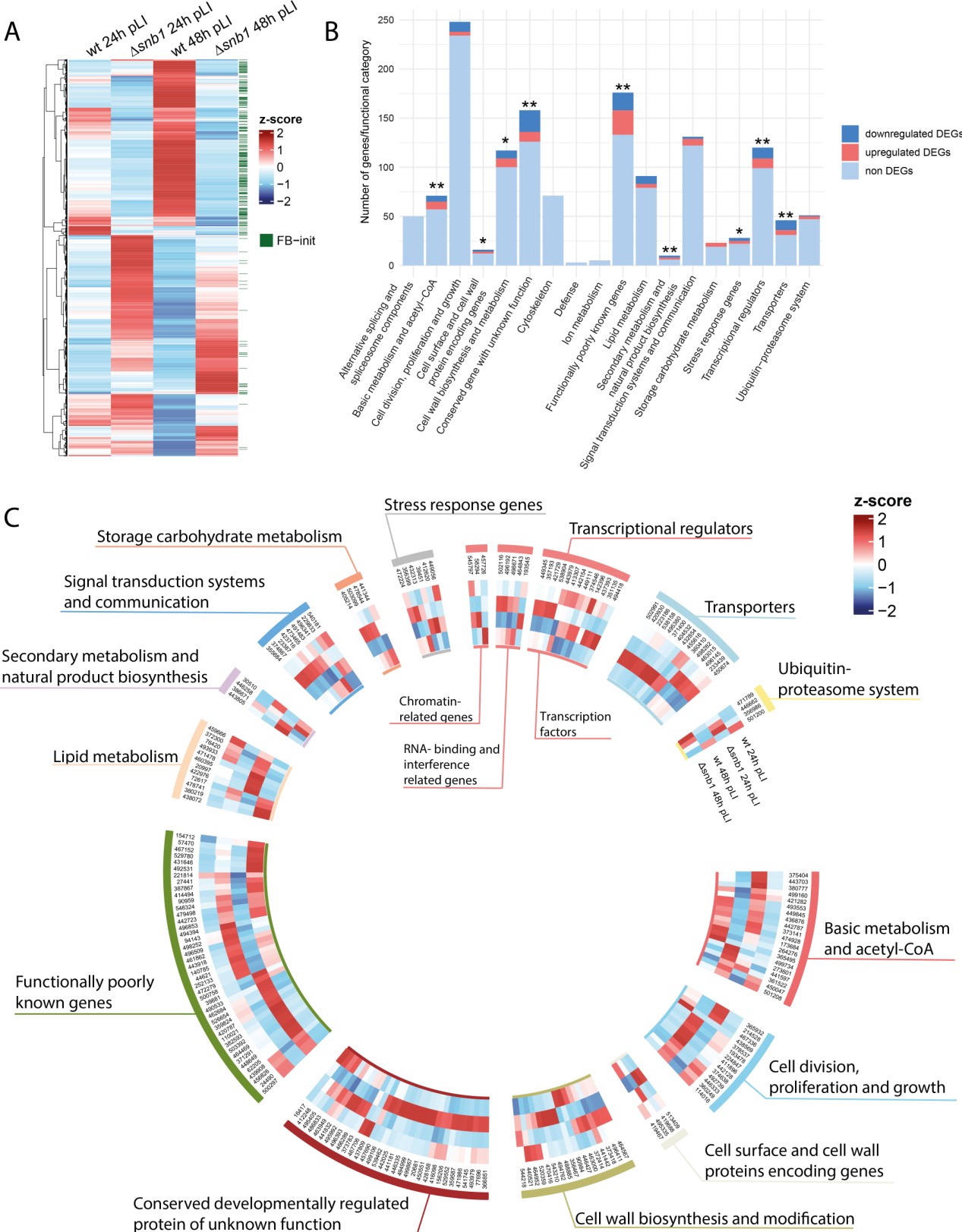

**FIG 5** Results of RNA-Seq analysis. (A) Heatmap of the 1,299 differentially expressed genes. (B) Functional categories of developmentally regulated genes according to Nagy *et al*. (2023). Asterisks indicate the significant enrichment of DEGs in a given category (one-tailed Fisher's exact test, *$P \leq 0.05$, **$P \leq 0.01$). (C) Heatmap displaying the normalized expression levels of differentially expressed genes (DEGs) within the 14 functional categories.

To better interpret the functions of DEGs, we classified them into 18 manually curated functional categories defined for conserved development-related genes, which cover 1,410 genes of *C. cinerea* (28). In this way, 205 DEGs were classified into fourteen functional categories (Fig. 5B and C; Table S2). Overall, Δ*snb1* DEGs are significantly enriched ($P = 1.004 \times 10^{-12}$, one-tailed Fisher's exact test) in the conserved functional categories, indicating that, DEGs include a considerable proportion of conserved genes and that overlaying DEGs with these functional categories provides a good basis to examine processes altered in the mutant. DEGs were significantly enriched in nine categories (Fig. 5B, one-tailed Fisher's exact test, $P \leqq 0.05$). The most strongly enriched categories ($P \leqq 0.01$, one-tailed Fisher's exact test) contain transcriptional regulators, transporters, secondary metabolism genes, basic metabolism-related genes, conserved unannotated genes, and functionally poorly known genes. Next, we briefly summarize the enriched functional categories and discuss *C. cinerea* DEGs belonging to these, including both conserved genes and those specific to *C. cinerea*.

### Basic metabolism and acetyl-CoA

This functional category includes several metabolic pathways that directly or indirectly converge on increased acetyl-CoA production, which is the central intermediate during the metabolism of non-fermentable carbon sources and is a building block of biological membranes (44). It was speculated that developmental regulation of genes in this category supplies acetyl-CoA for cell membrane assembly during rapid cell expansion in the growing fruiting body (28). Of the 20 DEGs in this category, five were downregulated in the Δ*snb1*. These changes in gene expression are predicted to primarily affect oxaloacetate catabolism. Oxaloacetate, which produces oxalate and acetate from oxaloacetate, is significantly downregulated (ID: 499734, $FC_{wt/\Delta snb1}$: 3.20), perhaps leading to a reduction in acetate and oxalate levels. The upregulation of oxalyl-CoA synthase (ID: 474928, $FC_{\Delta snb1/wt}$: 4.30) may further contribute to the reduction of oxalate levels. Conversely, oxalate decarboxylase (ID: 361522, $FC_{wt/\Delta snb1}$: 4.58), which converts oxalate to formate and $CO_2$, showed significant downregulation, as did carbonic anhydrase (ID: 450047, $FC_{wt/\Delta snb1}$: 2.49), an enzyme that catalyzes the hydration of $CO_2$ into bicarbonate and $H^+$ (45). These changes may alter oxalate production and intracellular pH homeostasis in the mutant.

Although we cannot definitively predict metabolite changes based on the expression dynamics of these genes alone, we hypothesize that the oxalate pathway and oxaloacetate synthesis are impaired in the mutant. Oxalic acid is a very common organic acid secreted by fungi (46) but its exact role remains unclear. Studies on wood decay have shown that oxalate can serve as a source of protons for the enzymatic and non-enzymatic hydrolysis of wood carbohydrates (47). In addition, oxalate can bind non-covalently to β-glucan and form a hydrogel in the extracellular matrix, which can influence the secretion of metabolites (48), which raises the possibility that these transcriptome changes influence cell wall functioning. Oxaloacetate is a precursor molecule of acetyl-CoA and other metabolites that are important for the biosynthesis of various compounds such as ergosterol and membrane lipids, which are essential for cell expansion. Therefore, the expression changes of genes related to basic metabolic systems may influence growth and cell wall dynamics in Δ*snb1*.

### Cell wall biosynthesis and modification

Cell wall biosynthesis and remodeling are probably among the most important processes in the development of fruiting bodies (7, 26). The fungal cell wall consists mainly of chitin, β−1,3-glucan and α−1,3-glucan, and other polysaccharides (49). In line with this, GO terms related to the fungal cell wall and chitin binding were enriched among downregulated DEGs (Fig. S5). We detected a total of 17 conserved DEGs in this category, which mainly comprise chitin and glucan remodeling genes. Generally, genes encoding chitin remodeling enzymes were downregulated, whereas those that code for chitin synthases and glucanases were upregulated in Δ*snb1*. Out of nine chitin

remodeling genes, four were downregulated ($FC_{wt/\Delta snb1}$: 2.76 to 2.11), all of which contain an active site of glycosyl hydrolase family 18. These include the endochitinase ChiEn3 (ID: 470416, $FC_{wt/\Delta snb1}$: 2.60), which was suggested to be involved in stipe cell wall modification (10). Downregulated genes included ID: 464952 and ID: 440521, which encode chitin deacetylases, a protein family that was previously reported to be involved in stipe development and elongation (11). Genes (ID: 372414, ID: 356467), encoding enzymes involved in chitin synthesis, were upregulated. Two genes with glycoside hydrolase 20 annotation (ID: 464567 and ID: 446427) were upregulated in the mutant; however, the function of these in development is currently unclear (28).

On the other hand, the three DEGs predicted to be involved in glucan remodeling were upregulated. These include a GH5 beta-glucanase, a GH3 member, and a member of the GH128 endoglucanase family. The latter has been first described in the fruiting bodies of *L. edodes* (50). Furthermore, two genes encoding enzymes with polysaccharide lyase domains were downregulated (ID: 494762 and ID: 543210, $FC_{wt/\Delta snb1}$: 2.27 and 5.91, respectively). Our knowledge of fungal lyases in relation to fungal development is limited, but they may have an impact by altering cell wall polysaccharides (26).

Taken together, the presence of chitinases and other enzymes that modify cell wall polysaccharide components suggests that processes related to cell wall synthesis and remodeling are affected in the mutant strain. The altered expression patterns of these genes could contribute to the spherical growth of Δ*snb1* fruiting bodies.

### Cell surface and cell wall protein-encoding genes

Several cell wall-associated gene groups, like lectins, hydrophobins, and cerato-platanins exhibited high expression differences in the Δ*snb1* mutant. We identified four DEGs in this functional group, however, because such genes evolve fast and did not form conserved orthogroups in the study that defined these categories (28), we decided to analyze all DEG genes in *C. cinerea* that are related to the cell surface.

Lectins are a highly variable group of genes, many of them encode secreted cell surface proteins that are known for their specific recognition and binding of various carbohydrate moieties, which makes them versatile in multiple biological roles, such as defense (51, 52). By combining MycoLec (53) and InterPro searches, we found a total of 95 lectins in the genome of *C. cinerea*, with 20 of these being DEG in the Δ*snb1* strain. Expression of ricin-B-like (beta-trefoil) lectins *ccl1* (ID: 456849) and *ccl2* (ID: 408852) increase more than 150-fold during primordium development in wt *C. cinerea* (52) but their expression levels in the Δ*snb1* strain did not show such an increase ($FC_{wt/\Delta snb1}$: 6.15 and −5.69, respectively). Similarly, two galectins, *cgl1* (ID: 473274) and *cgl2* (ID: 488611) were downregulated in Δ*snb1* ($FC_{wt/\Delta snb1}$: 4.69 and 2.94, respectively). Both galectins are involved in fruiting body formation: while *cgl2* is primarily expressed in hyphal knots, *cgl1* is expressed during fruiting body formation (54). A ricin-B lectin (ID: 442572) showed the highest upregulation in Δ*snb1* ($FC_{\Delta snb1/wt}$: 11.9). Numerous lectins show dynamic expression during fruiting body development (52, 55) and, as shown by this study, are altered in the Δ*snb1* strain. This underscores their role in sculpting fruiting bodies; however, whether they are part of the defensive arsenal or contribute in other ways remains to be established for each specific gene.

Hydrophobins can spontaneously assemble into an amphipathic monolayer to form a hydrophobic surface. Differentially expressed hydrophobins are highly enriched; of the 34 *C. cinerea* hydrophobins, 19 were DEGs in Δ*snb1* fruiting bodies, of which 11 were downregulated in the mutant. Among the downregulated DEG hydrophobins 8 were "FB-init" in the data set of Krizsán et al. (26). Hydrophobin up- or downregulation could indicate the lack of hydrophobic cell surface coating of mutant cells or, if the given hydrophobin gene is strongly tissue-specifically expressed, could also stem from the lack of certain tissues in the mutant strain.

Cerato-platanins are small, secreted, cysteine-rich proteins associated with the fungal cell wall (56). They were first described in the context of plant pathogenicity (57), however, based on their expression dynamics they are likely also associated

with development (26, 28). Three *C. cinerea* cerato-platanin genes showed significant downregulation in Δ*snb1*. Structurally, cerato-platanins show a double-Ψ-β-barrel (DPBB) topology of the RlpA superfamily, an architecture that is also typical to further proteins, like expansins (58, 59), certain endoglucanases (60), and virulence factors of plant pathogens (61, 62). An additional interesting example with this domain structure is the DEG ID: 421450, showing the highest upregulation (FC$_{Δsnb1/wt}$: 21.86) in the Δ*snb1* strain. The expression of this gene is generally very low during fruiting body development and encodes a secreted protein without further functional information (26). Cerato-platanins share structural and functional similarities with expansins, suggesting that some of them are involved in different stages of fungal growth and development and are possibly linked to the cell wall (26, 28, 63). This hypothesis is supported by previous studies on DPBB-containing effectors, which showed chitin-binding ability and suppression of chitin-triggered immunity (64).

Another group of DEGs worth highlighting encodes secreted proteins with a domain of unknown function 4360 (DUF4360). Five DUF4360-containing DEGs showed a remarkable upregulation in Δ*snb1* (e.g., ID: 116193, FC$_{Δsnb1/wt}$: 18.9), while one of them was downregulated (ID: 464561; FC$_{wt/Δsnb1}$: 6.98). These proteins have been described in certain plant pathogenic fungi as putative effectors (65). The literature on DUF4360-containing proteins in basidiomycetes is limited but one study reported mycorrhiza-induced small secreted proteins with DUF4360 in the ectomycorrhizae of *Hebeloma cylindrosporum* (66). The ortholog of this gene in *C. cinerea* (ID: 545612, FC$_{Δsnb1/wt}$: 17.69) was highly upregulated in Δ*snb1*.

A secreted protein-encoding gene, *PriA* (ID: 419688) is downregulated (FC$_{wt/Δsnb1}$: 8.22) in the Δ*snb1* strain. *PriA* has been studied in *L. edodes* with regard to fruiting body initiation and it might have a role in zinc homeostasis (32, 67). This gene is highly induced in primordia and young fruiting bodies of numerous basidiomycetes (26, 68, 69), and its downregulation in the mutant further underscores its importance.

## Secondary metabolism and natural product biosynthesis

The basidiomycete secondary metabolome is remarkably different in biogenetic origin and structure from the metabolites from ascomycetes (70). It includes polyketides, terpene, and amino acid derivatives from central metabolic pathways and primary metabolite pools (71, 72). Within this category, all DEGs belonged to the "FB-init" category (26), two being downregulated and two upregulated in the Δ*snb1* strain. Two sesquiterpene-producing enzymes showed differential expression: Cop2 (ID: 443805, FC$_{wt/Δsnb1}$: 7.64) was downregulated, while Cop4 (ID: 30510, FC$_{Δsnb1/wt}$: 3.53) was upregulated (73). In addition, a phenylalanine ammonia-lyase (ID: 386671, FC$_{wt/Δsnb1}$: 2.46) was downregulated, while a putative polyketide synthase (ID: 446258, FC$_{Δsnb1/wt}$: 6.61) was strongly upregulated in the Δ*snb1*. Terpene synthases are highly overrepresented in the genomes of Agaricomycetes compared to other fungi (26, 74) and may be involved in important functions of the fruiting body. The role of terpene synthases in fruiting is already well known in ascomycetes. For example, polyketide synthase genes in ascomycetes are essential for sexual development and fruiting body morphology. Their absence leads to immature fruiting bodies, while overexpression causes enlarged and malformed ones (75). The role of secondary metabolism in Basidiomycetes is less explored; nevertheless, the observed differential expression patterns in the mutant strain suggest that secondary metabolites may indeed play a role in this process.

## Transcriptional regulators

This category includes some of the most important genes that orchestrate the sequence of events in the developmental process, including genes related to chromatin regulation, RNA interference, those encoding RNA-binding proteins, and transcription factors (TFs) (16, 76). Of these, several TFs were significantly differently expressed in the Δ*snb1* strain. Of the 47 conserved TFs in this functional category, eight were downregulated and

five were upregulated. Five downregulated TFs were developmentally regulated during early developmental events ["FB-init" (26)]. The most downregulated TF is the ortholog of *fst1* (ID: 437393, $FC_{wt/\Delta snb1}$: 7.35) of *S. commune*, the deletion of which leads to arrested fruiting body development (18). The HMG-box containing *exp1* (ID: 421729) was upregulated in Δ*snb1* fruiting bodies ($FC_{\Delta snb1/wt}$: 3.26). *Exp1* was suggested to be required for pileus expansion and autolysis of *C. cinerea*, controlling the final phase of fruiting-body morphogenesis (77). Three out of five members of the transcriptional regulatory Velvet-complex (IDs: 355684, 374867, 496341; $FC_{\Delta snb1/wt}$: 2.09–3.73) showed upregulation in the mutant. Velvet-complex genes are known to be upregulated in the early stages of fruiting body development in several basidiomycetes, suggesting a possible role in their sexual development (25).

The ortholog of the *Schizosaccharomyces pombe* RNA-binding protein *mei2* (ID: 193545, $FC_{wt/\Delta snb1}$: 3.96) is downregulated in the mutant. *Mei2* is a conserved regulator of meiosis that protects meiosis-specific transcripts from degradation (25). Similar to other meiosis-related genes typically upregulated in cap structures (68), *mei2* showed increased expression in primordia and young fruiting bodies in several basidiomycetes (25).

On the other hand, of the 48 chromatin-related genes within this category, we identified only two downregulated DEGs in the Δ*snb1,* both encoding methyltransferases. *lch1* (ID: 545797, $FC_{wt/\Delta snb1}$: 13.34) (17) and ID: 501966, $FC_{wt/\Delta snb1}$: 4.84) (15), both putatively involved in chromatin regulation. *lch1* mutation in *C. cinerea* resulted in poorly differentiated fruiting bodies, resembling *snb1* mutants (17).

### Transporters

Conserved transporter-encoding genes were strongly enriched among Δ*snb1* DEGs, which was also evident from the GO enrichment analysis (Fig. S3). Of the 15 DEG transporters, 10 were downregulated in the mutant. Four downregulated conserved DEGs belong to the Major Facilitator Superfamily (MFS), with ID: 371400 being the most downregulated ($FC_{wt/\Delta snb1}$: 10.16). MFS is one of the largest transporter families, with diverse functions (78). A zinc-regulated transceptor (ID: 455616, $FC_{wt/\Delta snb1}$: 3.32) and a urea/ammonium transporter (ID: 538158, $FC_{wt/\Delta snb1}$: 3.91) were also downregulated in the mutant. These transporters may fulfill functions in fruiting bodies by facilitating the movement of various types of substances, including ions, signaling molecules, osmolytes, and nutrients. However, due to their rapid evolution and the lack of more precise functional data, it is difficult to infer their functions with more specificity.

### Functionally poorly known genes

This category mainly includes genes with predicted functions that cannot be easily interpreted (28). We detected 43 DEGs in this category, of which 18 are downregulated and 25 are upregulated. Among the most downregulated DEGs, we found a gene encoding nitronate monooxygenase (ID: 110021, $FC_{wt/\Delta snb1}$: 11.75), which has been described mainly in pathogenic ascomycetes, where it catalyzes the denitrification of nitroalkanes associated with nitro-oxidative stress (79). In addition, two cytochrome P450 encoding genes (ID: 456826 and ID: 24490; $FC_{wt/\Delta snb1}$: 9.47 and $FC_{wt/\Delta snb1}$: 4.54, respectively) were also downregulated in the Δ*snb1*. These enzymes are involved in the oxidative, peroxidative, and reductive metabolism of various compounds. Deletion of another cytochrome P450 gene, *eln2* in *C. cinerea* resulted in the formation of mature fruiting bodies with short stipes (80), suggesting that other cytochrome P450 enzymes may also contribute to cell morphogenesis and tissue organization.

### Genes related to reactive oxygen species (ROS)

ROS generated either externally or as a result of normal metabolism can be harmful to various cellular components; therefore, to balance the flux and redox reactions within cells, fungi employ complex stress response pathways (81). Genes encoding enzymes that protect against ROS, are differentially regulated in the Δ*snb1* strain.

Within this functional category, thioredoxin genes (ID: 412620 and 39451, $FC_{wt/\Delta snb1}$: 5.25 and 2.42) are downregulated, while a catalase (ID: 472224, $FC_{\Delta snb1/wt}$: 3.13) is upregulated. In *Aspergillus nidulans,* the deletion of its thioredoxin gene led to increased catalase activity, and the mutant strain showed decreased growth and inability to differentiate reproductive structures (82). Antioxidant-producing enzymes showing upregulation in the Δ*snb1* strain included Egt1, ergothionine biosynthesis protein 1 (ID: 356399, $FC_{\Delta snb1/wt}$: 2.57) and Alo1, a D-arabinono-1,4-lactone oxidase (ID: 432313, $FC_{\Delta snb1/wt}$: 4.61). Further related, non-conserved genes, involved in defense against oxidative damage showed upregulation in the mutant, like a superoxide-dismutase (ID: 420887, $FC_{\Delta snb1/wt}$: 3.10) and six out of the seven differentially expressed gluta-thione-*S*-transferase genes ($FC_{\Delta snb1/wt}$: 2.08–3.92). In eukaryotic microorganisms, ROS are generally considered as detrimental by-products of aerobic metabolism, however, increasing evidence suggests that their presence in specific concentrations contributes to important physiological roles and triggers cell differentiation in response to oxidative stress (83–85). ROS generated by NADPH oxidases (NOX) can be involved in signaling processes driving cellular differentiation (86). However, in our data, the expression of NOX genes was not found to be altered in the mutants. Nevertheless, the elevated expression level of ROS scavenging enzyme encoding genes in the Δ*snb1* may lead to a shift in the cellular oxidative environment, which could indirectly play a role in the arrest of differentiation observed in the Δ*snb1* strain.

## Protease inhibitors

Fungal protease inhibitors are important regulators of proteases; therefore, they play essential physiological roles in fungal cells, such as resource recycling and defense (87). Cap-specific serine protease inhibitors belonging to the MEROPS family I66 are known as mycospins, which, in *C. cinerea*, have been named cospins (52). In Δ*snb1,* two paralogous cospins were downregulated (ID: 497809 and ID: 367957, $FC_{wt/\Delta snb1}$: 6.73 and 3.69, respectively). Both were reported to be upregulated during fruiting body initiation ("FB-init") and enriched in the cap tissue (25, 26). Cospins were initially characterized as having a defensive function against predators and parasites (88). However, it is possible that they serve a dual biological purpose by also controlling the activity of internal serine proteases that play a role in the development of fruiting bodies (87).

## Isoprenylcysteine O-methyltransferases

One of the strongest signals in the GO analysis of downregulated genes (Fig. S5) was related to two isoprenylcysteine *O*-methyltransferase encoding genes (ID: 354499, ID: 445523, $FC_{wt/\Delta snb1}$: 6.27 and 9.71, respectively). These are responsible for carboxyl methylation of proteins that terminate with a CaaX motif, such as mating pheromones and small GTPases (89, 90), which are downregulated in the mutant. We speculate these could be involved in posttranslational cleavage of pheromone precursors or other developmentally relevant polypeptides.

## Wax synthases

Wax synthases are involved in the production of wax esters in plants, which form a hydrophobic cuticle layer that serves as a protective barrier against biotic and abiotic stresses (91), and may play a similar role in fungi. Of the eight *C. cinerea* wax synthase genes, three were significantly downregulated in the Δ*snb1* mutant [as an example: ID: 478454 with the largest difference ($FC_{wt/\Delta snb1}$: 6.49) in expression].

Taken together, the RNA-Seq data presented here for two developmental stages of wt and Δ*snb1* strains suggest that the lack of developmental patterning that results from the disruption of *snb1* affects numerous molecular pathways and SNB1 might have a central role in fruiting body development. Our analyses of the gene expression data highlighted an enrichment of genes that, in previous studies, were shown to be upregulated during

early development, when cell differentiation takes place. This is underscored by the preponderance of tissue specifically expressed genes among Δ*snb1* DEGs, indicating that, in line with the Δ*snb1* phenotype, our RNA-Seq analysis has probably successfully captured genes involved in the differentiation of fruiting body cells. We envision these genes can form the basis of further studies of cellular differentiation in *C. cinerea*, one of the key features of complex multicellularity. In the future, further reverse genetics studies, especially of regulatory genes, could test the functions of these genes to provide further resolution on the genetic circuits that orchestrate differentiation in mushroom fruiting bodies.

## DISCUSSION

In this paper, we characterized *snb1* of *C. cinerea*, a member of a novel, conserved gene family that influences development and cell differentiation in fruiting bodies of Agaricomycetes. We identified *snb1* in previous developmental transcriptomic data (26, 28), based on the conservation of its upregulation upon the initiation of fruiting body development. *snb1* encodes a protein with a domain of unknown function, which, in conjunction with its expression patterns, made it an exciting candidate for functional characterization. In *C. cinerea*, the deletion of the *snb1* gene resulted in globose, snowball-like fruiting bodies that failed to properly differentiate cap and stipe tissues and remained stunted (Fig. 1).

To characterize *snb1* using reverse genetics, we applied an RNP-complex-based CRISPR/Cas9 system (34), and obtained three independent mutants, displaying the same phenotype. While we could not observe any significant changes in the growth of vegetative mycelia, the development of the fruiting bodies showed severe deficiencies, reminiscent of snowballs, which were unable to form a fully structured cap, stipe, or gills, resulting in a lack of sporulation. Although rudimentary cap-like structures emerged later, these did not develop further into recognizable cap initials. To the best of our knowledge, similar phenotypes have not been identified before in *C. cinerea* or other agaricomycetes. The *ichijiku* mutant, which is caused by a point mutation in the putative chromatin remodeling gene *ich1* (17) exhibits some similarity, as both mutants fail to develop a differentiated pileus at the apex of the primordial shaft and a dark stipe. However, in the *snb1* mutant, the rudimentary veil appears at later stages (4 days post-induction), and the primordia remain spherical, whereas the *ichijiku* mutants elongate vertically. Together, phenotypic data indicate that *snb1* may be involved in the differentiation of *C. cinerea* fruiting bodies and based on the conservation of its expression patterns, probably in that of other agaricomycetes too.

Phylogenetic analyses identified orthologs and homologs of *snb1* across the Agaricomycetes, indicating broad conservation among fruiting body-forming basidiomycetes (but not in fruiting body-forming ascomycetes). Despite this conservation, the encoded protein does not contain any known conserved domain with a known function. Such proteins are widespread in fungal genomes and often correspond to novel gene families that are yet to be functionally characterized. Unannotated genes in fungal genomes are frequently highly variable in sequence and encode effectors or small secreted proteins. Examples of recently characterized such gene families include mycorrhiza-induced small secreted proteins (92), a short *Cryptococcus neoformans* peptide related to quorum sensing (93), effector-like Ssp1 from *Pleurotus ostreatus* (94), or a cell-surface effector complex from *Ustilago* (95), among others. However, not all genes with unknown function lack conservation. Several of them are conserved, such as the septation-related *spc* genes (33) or the recently reported pathogenicity-induced small secreted proteins of *Armillaria* (96). In the context of agaricomycetes' fruiting bodies, a recent study identified 158 orthogroups of conserved genes without known function that showed developmentally dynamic expression patterns (28). *snb1* was one of these, and our results indicate that the conservation of developmental expression was an accurate predictor of a role in fruiting body development. We interpret these results as evidence these conserved genes can be a rich source of developmental genes and

potentially novel functionalities in mushroom-forming fungi. More research is needed to uncover their role in fruiting body development and other life-history traits of fungi.

While the phenotype, phylogenetic conservation, and biophysical properties of SNB1 are straightforward, it is much harder to speculate about its mechanistic function in fungal cells. Structural analyses of the polypeptide indicated that SNB1 is probably a cell membrane protein with seven transmembrane helices (Fig. 4A and B). Based on this, it could function as a transporter, channel, or receptor. However, typical transporters and ion channels generally have more transmembrane domains. The 7 transmembrane domain structure is typical for G-protein-coupled receptors, which, however, have a strict classification system, to which SNB1 does not conform. It is possible that SNB1 fulfills one of these, or yet other roles in the cell membrane but clarifying its mechanistic role in development will require more research.

While the exact function of *snb1* remains elusive, we were able to take advantage of the lack of proper tissue differentiation in the Δ*snb1* strain to obtain insights into the genes underpinning the development of tissue types in *C. cinerea*. We profiled gene expression in two developmental stages in which differentiation happens and in which *snb1* is upregulated in wt *C. cinerea* and identified 1,299 significantly differentially expressed genes. Downregulated genes were enriched in ones that, in previous wt data expression (26) show a marked upregulation in secondary hyphal knots and stage 1 primordia as well as displayed tissue-specific expression patterns. These included several previously described fruiting-related genes, such as hydrophobins, lectins (*ccl1-2, cgl1-2*), cell wall synthesis and remodeling genes, transcriptional regulators (*exp1* and *fst1* transcription factors, *ich1-2*) and protease-inhibitors involved in defense, among others. We also identified genes that recurrently appear in fruiting body transcriptomes but have not yet been verified to participate in the morphogenetic process. These include several transcription factors which, as regulators, may be involved in orchestrating differentiation processes in the fruiting body but also cerato-platanins, wax synthases, secondary metabolism genes, or those involved in oxalate metabolism. We anticipate that these genes will be worthy targets of further reverse genetics approaches aimed at uncovering key elements and mechanisms of fruiting body development and differentiation.

It has been reported that LiCl inhibits fruiting body development in a concentration-dependent manner, perhaps by acting on GSK-3, a conserved kinase among eukaryotes that controls various biological activities, such as glycogen metabolism (37, 97–99). Our study confirmed the inhibitory effect and revealed that, in the presence of LiCl at concentrations that arrested the wt strain at the hyphal knot stage, the Δ*snb1* strain was still capable of producing more advanced, elongated structures (Fig. 3). These observations indicate that the Δ*snb1* mutant is less sensitive to the inhibitory effect of LiCl, although the mechanistic details of this difference in sensitivity are not known currently. If LiCl indeed acts by inhibiting glycogen metabolism (GSK-3), then these data might indicate that glycogen metabolism is less active, or less required for the growth and developmental progression of Δ*snb1* fruiting bodies. However, clarifying this will require further research.

In summary, this study reported a novel gene family in Agaricomycetes and demonstrated in *C. cinerea* that one of its members, *snb1* is involved in fruiting body development. These data add to our knowledge of development-related genes in mushroom-forming fungi. *snb1* is a gene that encodes a protein with no known functional domains. This offered few clues as to how SNB1 might mechanistically influence development. Nevertheless, such genes and the encoded proteins might be involved in novel pathways, or unknown subcellular structures, which are worth examining in more detail, for example by mapping genetic and physical interactions in the cells. Eventually, characterizing a more complete array of functionally poorly known genes will allow us to uncover the molecular complexity of the genetic program that governs fruiting body development, one of the most complex developmental events in fungi.

## MATERIALS AND METHODS

### Bioinformatic analyses of *snb1*

We used the JGI ProteinID accession numbers of the *C. cinerea AmutBmut (#326)* strain (https://mycocosm.jgi.doe.gov/Copci_AmutBmut1/Copci_AmutBmut1.home.html). In addition, Table S2 also includes the compiled ProteinID accession numbers of the Okayama 7 (#130) strain.

### Phylogenetic analysis of SNB1

To assess SNB1 conservation, we used a previously published data set and species tree containing 109 species mainly from Agaricomycotina (22). An OrthoFinder2 (100) clustering was applied to all protein-coding genes of these 109 species with the default parameters but a fixed species tree topology. We manually checked species in which Orthofinder did not find orthologs, using reciprocal best BLAST hits. In the species (Sissu1) *Sistotremastrum suecicum* (Sissni1), *Sistotremastrum niveocremeum,* and (Sclci1) *Scleroderma citrinum,* we found SNB1 homologs using BLAST, independently from the abovementioned clustering analysis. Multiple sequence alignment on the cluster containing SNB1 was inferred using the algorithm MAFFT v6.864b L-INSI-I (101) and trimmed using TrimAL v.1.2 (-gt 0.2) (102). Maximum likelihood gene tree inference (ML) was performed using the LG + G model of sequence evolution and branch robustness was assessed using 1,000 ultrafast bootstrap replicates and the SH-aLRT support measure in IQ-TREE v1.6.12 (103). Rooting and gene tree/species tree reconciliation were performed with NOTUNG v2.9 (104) using an edge weight threshold of 80. Finally, COMPARE (105, 106) https://github.com/zsmerenyi/compaRe) was used to identify and visualize orthogroups within the gene tree and to assess the duplication and loss history of the cluster.

### Functional annotations

Conserved domains in proteins were detected using InterProScan-5.61–93.0 (107). Enrichment analysis on GO categories was carried out using the R package topGO 2.44.0 (108). WoLFPSORT (109) was used to predict subcellular localization. DeepTMHMM [version: 1.0.24; (110)] was used to determine the number and orientation of transmembrane helices and AlphaFold [(111, 112); version 2022–11-01] was used to predict tertiary structure.

### Strains and culture media

The *p*-aminobenzoic acid (PABA) auxotrophic homokaryotic *C. cinerea* AmutBmut1 *pab1-1* #326 strain was cultured and maintained in a YMG medium. The composition of media and buffers used in the transformation (YMG, minimal medium, regeneration medium, top agar MM- and MMC buffers, and PEG/CaCl$_2$ solution) followed the description of Dörnte and Kües (2012) (113). To obtain oidia, *C. cinerea* was cultured on YMG at 37°C under continuous light for 6 days. The mutant *C. cinerea* strains with complemented *pab1* gene were kept in a minimal medium (MM) without PABA.

All the shown phenotypes and the further experiments, including the gene complementation and sampling for RNA-Seq analysis were performed on the same strains (wt and *Δsnb1*/II. strain).

### Design of sgRNAs and RNP-complex assembly

We designed sgRNA using the software sgRNAcas9 (114) using the exon sequences of the *snb1* gene (protein ID: 493979). The reference genome (ID: GeneModels_FrozenGene-Catalog_20160912.fasta) was downloaded from the JGI database (115). The protospacer and PAM sequence were (N)20-NGG, and the number of mismatches of the protospacer was minimized when mapping to the whole reference genome of *C. cinerea*. The selected

sgRNA (AUGCCGUGGUACUCUGGGAU) had minimal predicted off-target effects in the *C. cinerea* genome, and its predicted Cas9 cleavage site was located on 17 bp in exon 7. The trans-activating CRISPR RNA (tracrRNA) and crRNA were commercially synthesized (IDT, USA) for the *in vitro*-assembled RNP complexes. The *in vitro*-assembled RNP complexes were obtained by mixing 1.2 µL each of 20 µM crRNA and tracrRNA with 9.6 µL duplex buffer (IDT, USA), following the manufacturer's instructions. The mixture was incubated at 95°C for 5 minutes and cooled to room temperature for 5 minutes. Next, 0.5 µL duplex buffer, 1.5 µL Cas9 working buffer [based on Liang *et* al., 2018(116)], and 1 µg TrueCut HiFi Cas9 enzyme (Invitrogen, A50575) were added to the mixture, and the reaction was incubated at 37°C for 15 minutes. The assembled RNP-complex was stored on ice until use. Then, 15 µL of RNP mix was used in protoplast transformation.

## Repair template preparation

To obtain the linear repair template, we constructed the KO_493979_pab1_pUC19 plasmid, in which the backbone was a pUC19 vector and consisted of the positive selection marker gene *pab1* surrounded by 1000 bp long homologous arms (113), which were amplified with overlapping Gibson primers starting 10 bp away from the expected Cas9 cleavage site. The positive selection marker gene *pab1* with its own promoter and terminator region was amplified from the pMA412 vector (117) and flanked by the upstream and downstream homologous arms. Amplicons were purified using the QIAquick PCR Purification Kit (Qiagene) and the KO_493979_pab1_pUC19 plasmid was constructed using an enhanced Gibson Assembly Mix following the protocol of Rabe and Cepko (2020) (118). The repair template was amplified with primers of pUC19_checking_fwd and β-tubT_(pUC19)_rev. All fragments used in the vector construction and the homologous repair template were amplified using Phusion Plus High-Fidelity Polymerase (Thermo Scientific), following the manufacturer's instructions. The primers used in this study are listed in Table S1.

## Protoplast generation from *C. cinerea* mycelia

*C. cinerea* was precultured on YMG at 37°C and under light for 6 days to produce oidia. Oidia were harvested by gently brushing the surface of the culture after adding 5 mL of distilled water. The oidium suspension was inoculated into 50 mL of YMG and incubated overnight at 37°C with shaking at 120 rpm. Germinated oidia were collected and centrifuged at 2,600 × *g* for 5 minutes. The supernatant was discarded and mycelia were resuspended in 40 mL of sterile MM buffer containing 2% VinoTaste Pro digestive enzyme mix (Novozymes) and gently shaken (100 rpm) at 37°C for 3–4 hours. The digestion process was stopped by adding $CaCl_2$ at a final concentration of 25 mM. Protoplasts were filtered using a 40 µm pore size cell strainer (VWR) and the filtrate was centrifuged at 640 × *g* for 10 minutes at 4°C. The supernatant was discarded and the protoplasts were washed by careful resuspension in 5 mL MMC buffer and collected as described above. The protoplasts were carefully resuspended in 100 µL MMC buffer per transformation and kept on ice until use.

## Transformation of *C. cinerea*

Protoplasts were transformed with a DNA repair template and RNP-complex using a modified PEG/$CaCl_2$-mediated transformation protocol as described by Dörnte and Kües (2012) (113). The transformation mixture contained 25 µL PEG/$CaCl_2$ solution, approximately 1 µg DNA repair template, and 15 µL RNP complex for each 100 µl (~ $10^6$–$10^7$) protoplast suspension. To increase the permeability of the cell membrane and facilitate the entry of the RNP complex into the protoplasts, the surfactant Triton X-100 was added to the protoplast suspension at a final concentration of 0.006% (119). The transformation mixture was incubated on ice for 30 minutes. Then a heat shock step was performed by adding 500 µL PEG/$CaCl_2$ to the transformation mixture and incubating it at room temperature for 10 minutes. The transformed protoplasts were carefully suspended in

a 5 ml MM buffer and then mixed with 35 mL top agar (40°C) before 10–10 mL of the mixture was layered onto four bottom agar plates containing regeneration medium without PABA. The regeneration plates were incubated at 37°C and the first colonies appeared after 3–4 days. The colonies were then transferred to MM agar plates without PABA for further analysis.

For complementation experiments, we transformed the protoplasted Δ*snb1* oidia with a complementation vector, which consisted of the *snb1* gene with its 1,000 bp homologous arms and a hygromycin resistance cassette. The transformed cells were spread onto plates with 25 mL regeneration bottom agar without PABA and hygromycin B and kept at 28°C overnight. Then, 5 mL 600 µg/mL hygromycin B containing top agar was layered on each plate. After incubation for several days at 37°C, the colonies were transferred to MM agar plates without PABA and hygromycin.

## Fungal colony PCR

To confirm gene disruption, a microwave-assisted colony PCR technique was used (120), in which a pinhead-sized mycelium sample was transferred to 100 µL sterile distilled water, then microwaved at 700 W for 1 minute, briefly vortexed and microwaved again at 700 W for 1 minute, and then stored at −20°C for at least 10 minutes. Prior to PCR, the thawed samples were centrifuged to pellet mycelia and the supernatant was used as a template for colony PCR using Phire Green Master Mix (Thermo Scientific) and checking primers, according to the manufacturer's instructions.

The checking primers were designed to be 100 bp away from the end of the homologous arms; hence, the expected amplicon size of the mutants was longer than that of the wt colonies due to the incorporation of the *pab1* selection marker gene or the whole linear repair template into the RNP complex-mediated cleavage site. The used primers are listed in Table S1.

To isolate pure strains from the obtained transformants, we initially cultured them in YMG medium under constant light at 37°C for 5 days. We then harvested the oidia from the mycelial surface using a 0.01% Tween-20 solution. Then, the oidium suspension was diluted and inoculated onto minimal media to separate all the strains.

## Synchronized induction of fruiting

To obtain fruiting bodies from the same developmental stages, we followed a previously described method (24). Briefly, *C. cinerea* strains (d = 5 mm mycelial plugs) were inoculated in the center of YMG agar plates with half glucose content (0.2% wt/vol) and incubated for 6.5 days at 28°C in the dark. Once the agar plate was almost completely colonized by mycelium, we initiated fruiting body development by exposing the colony to white light for 2 h, then placing it back in the dark for 24 h. This was followed by a 12-h light/12-h dark period at 28°C.

## Lithium-chloride treatment

To investigate the effect of LiCl on fruiting body development, YMG with half glucose content was adjusted to different LiCl concentrations (0 g/L, 0.5 g/L, 1 g/L, 1.5 g/L, 2 g/L). The strains were cultured, and fruiting was induced synchronously as described above. Five replicates were used for each LiCl concentration and strain.

## Mycelial growth rate determination on complete and C- or N-deprived media

To determine the mycelial growth rate on agar plates, we used complete YMG, complete synthetic Fries media (35), and Fries medium without carbon or nitrogen sources. To induce carbon- or nitrogen-starvation, glucose or ammonia-tartrate and asparagine were omitted from the Fries medium, respectively. All the Fries media were supplemented with PABA. Mycelial growth was monitored daily by marking the boundaries of the colonies on the bottom of the plates following inoculation with a 5 mm diameter plug

of *C. cinerea,* which was previously grown on a minimal medium at 28°C for 6 days in the dark. A total of six replicates were examined for each experimental condition. Mycelial growth was monitored by measuring the colony diameter. For statistical analysis of the differences between mutant and wt strains' mycelial growth rate means on different media we used a two-way analysis of variance (ANOVA) implemented in the software JASP (version 0.17.1) (121).

## Mycelial weight measurement

Wet and lyophilized dry mycelial weights of the three independent *snb1* knock-out mutant strains and the wt were determined after growing on sterile cellophane layered on complete YMG agar plate (d = 5 mm mycelium) at 28°C for 6 days in the dark. For statistical analysis of the differences between mutant and wt strains' mycelial weight means, we used one-way ANOVA implemented in the software JASP (version 0.17.1) (121).

## Sampling and sample preparation for RNA-Seq and data evaluation

To perform the RNA-Seq analysis, we obtained secondary hyphal knots and stage 1 primordia from synchronized colonies. Fungal samples were collected at 24 h and 48 h pLI. Samples were fixed with Farmer's solution (75% absolute ethanol and 25% glacial acetic acid) with 1% 2-mercaptoethanol. 10 mL of Farmer's solution was applied per colony followed by a 30-minute infiltration using a vacuum pump. Secondary hyphal knots and primordia were collected individually. Three biological replicates of each sample type were stored at –80°C until RNA extraction. Before RNA extraction, samples were washed three times with 1 mL of diethyl pyrocarbonate-treated distilled water and then homogenized with micropestles in liquid nitrogen, and total RNA was extracted by using the Quick-RNA Mini-Prep Kit (Zymo Research) following the manufacturer's instructions. RNA integrity was determined via gel electrophoresis and the concentration was quantified using Nanodrop. Strand-specific cDNA libraries were constructed from poly(A)-captured RNA using the Illumina TruSeq Stranded RNA-Seq library preparation kit and sequenced on the Illumina Novaseq 6000 (S4) PE150 with (39.7–51.9)/2 million read pairs per sample at Novogene (UK).

Raw reads were analyzed as follows. To remove adaptors, ambiguous nucleotides, and low-quality read end parts, reads were trimmed using bbduk.sh (part of BBMap/BBTools; http://sourceforge.net/projects/bbmap/) with the following parameters: trimq = 30, minlen = 50. A two-pass STAR alignment (122) was performed against the reference genome with the same parameters as in our previous study (22) yielding 13.8–18.3 million feature-assigned fragments per sample. Expression values were calculated as fragments per kilobase of transcript per million mapped reads (FPKM). Differentially expressed genes were identified using cutoff criteria where the log fold-change (LogFC) had to be less than –1 or greater than 1, and the adjusted p-value (Benjamini-Hochberg procedure) had to be less than or equal to 0.05. Genes with FPKM values below five in all samples were excluded from the analysis.

The data set of Krizsán et al. (2019) (26) was reevaluated to identify genes exhibiting tissue-enriched expression, using the R package 'TissueEnrich' (123) with the default parameters, except the threshold which was modified to four-fold-change.

The enrichment of functional categories was determined by a one-tailed Fisher's exact (hypergeometric) test using the phyper() function in R.

## ACKNOWLEDGMENTS

This research was funded by the Momentum Program of the Hungarian Academy of Sciences (LP2019-13/2019) and the National Research Development and Innovation Office (Grant No. OTKA 142188). Prepared with the support of the Doctoral Student Scholarship Program of the Co-operative Doctoral Program (KDP-17-4/PALY-2021) of the

Ministry of Innovation and Technology financed from the National Research, Development and Innovation Fund.

C.F., L.G.N., and L.G. conceived the study. C.F. generated mutants, performed the experiments, analyzed the data, and prepared figures. A.C., H.W., M.V., Z.H., and X.B.L. helped with wet-lab experiments, including CRISPR/Cas9 of *C. cinerea*. Z.M., C.F., B.B., and B.H. analyzed the bioinformatic data, C.F., L.G.N., L.G., and Z.M. wrote the paper. All authors have read and commented on the manuscript.

## AUTHOR AFFILIATIONS

[1]Synthetic and Systems Biology Unit, Institute of Biochemistry, HUN-REN Biological Research Center, Szeged, Hungary
[2]Doctoral School of Biology, Faculty of Science and Informatics, University of Szeged, Szeged, Hungary
[3]Department of Biotechnology, Faculty of Science and Informatics, University of Szeged, Szeged, Hungary

## AUTHOR ORCIDs

Csenge Földi  http://orcid.org/0000-0002-9982-0679
László Galgóczy  http://orcid.org/0000-0002-6976-8910
László G. Nagy  http://orcid.org/0000-0002-4102-8566

## FUNDING

| Funder | Grant(s) | Author(s) |
| --- | --- | --- |
| Hungarian Academy of Sciences | LP2019-13/2019 | László G. Nagy |
| Ministry of Innovation and Technology (Hungary) | KDP-17-4/PALY-2021 | Csenge Földi |
| National Research Development and Innovation Office (Hungary) | OTKA 142188 | László G. Nagy |

## AUTHOR CONTRIBUTIONS

Csenge Földi, Conceptualization, Data curation, Formal analysis, Methodology, Validation, Visualization, Writing – original draft, Writing – review and editing | Zsolt Merényi, Data curation, Formal analysis, Visualization, Writing – original draft | Bálint Balázs, Data curation, Formal analysis | Árpád Csernetics, Methodology | Nikolett Miklovics, Methodology | Hongli Wu, Methodology | Botond Hegedüs, Data curation, Formal analysis | Máté Virágh, Methodology | Zhihao Hou, Methodology | Xiao-Bin Liu, Methodology | László Galgóczy, Conceptualization, Supervision, Writing – original draft, Writing – review and editing | László G. Nagy, Conceptualization, Data curation, Funding acquisition, Investigation, Supervision, Writing – original draft, Writing – review and editing

## DATA AVAILABILITY

Short-read data associated with the study have been deposited in NCBI archive GEO under the accession number GSE245082.

## ADDITIONAL FILES

The following material is available online.

### Supplemental Material

**Supplemental Figures (mSystems01208-23-S0001.pdf).** Figures S1 to S7 and captions for the supplemental tables.
**Table S1 (mSystems01208-23-S0002.xlsx).** List of primers.

**Table S2 (mSystems01208-23-S0003.xlsx).** Transcriptomic data of differentially expressed genes.

**Table S3 (mSystems01208-23-S0004.xlsx).** GO enrichment analysis.

**Table S4 (mSystems01208-23-S0005.xlsx).** Mapping statistics for Δ*snb1* and wild type.

## Open Peer Review

**PEER REVIEW HISTORY (review-history.pdf).** An accounting of the reviewer comments and feedback.

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
