## [Reviewer comments · mSystems]

Snowball: a novel gene family required for developmental patterning of fruiting bodies of mushroom-forming fungi (Agaricomycetes)

Csenge Földi, Zsolt Merényi, Balázs Bálint, Árpád Csernetics, Nikolett Miklovics, Hongli Wu, Botond Hegedüs, Máté Virágh, Zhihao Hou, Xiao-Bin Liu, László Galgóczy, and László Nagy

Corresponding Author(s): László Nagy, Szegedi Biológiai Kutatóközpontja

Review Timeline:

Submission Date:	November 10, 2023
Editorial Decision:	December 18, 2023
Revision Received:	January 6, 2024
Accepted:	January 10, 2024

Editor: Benjamin Wolfe

Reviewer(s): Disclosure of reviewer identity is with reference to reviewer comments included in decision letter(s). The following individuals involved in review of your submission have agreed to reveal their identity: Yoichi Honda (Reviewer #3)

Transaction Report:

DOI: <https://doi.org/10.1128/msystems.01208-23>

Re: mSystems01208-23 (Snowball: a novel gene family required for developmental patterning in fruiting bodies of mushroom-forming fungi (Agaricomycetes))

Dear Dr. László Nagy:

Thank you for the privilege of reviewing your work. Three experts have reviewed your work and have suggested a variety of modifications that will help improve the clarity of the manuscript.

Revision Guidelines

Sincerely,
Benjamin Wolfe
Senior Editor
mSystems

Reviewer #1 (Comments for the Author):

Földi and co-authors have identified the gene snowball (snb1) as an important gene in the development of the mushroom-forming fungus *Coprinopsis cinerea*. It is considered unannotated and was primarily selected based on its expression profile in previously published data sets, as well as strong conservation in other mushroom-forming fungi. A gene deletion of snb1 resulted in stunted development. Next, RNA-Seq was performed to identify genes downstream of snb1. In general, the research appears to have been performed correctly. I have a few minor comments. The authors consider the gene/protein snowball as 'unannotated', even though they report that it has 7 transmembrane domains

and even a PFAM domain. Although the PFAM domain is essentially a domain of unknown function, I still don't consider this protein to be completely unannotated. A better term could be a putative transmembrane protein of unknown function, or something like that.

In my opinion the RNA-Seq dataset was not very useful to answer questions about the function of snowball. The authors carefully avoid stating that the transcriptomic changes are directly due to the deletion of snowball, which is good. Instead, they take the altered morphology as an opportunity to study genes involved in development. In principle that is fine, but in retrospect it would have been better to do RNA-Seq on a timepoint when WT and delta-snb1 first diverged in development (or even a bit before), in order to capture the early responses to the deletion. Over half the Results is dedicated to this RNA-Seq analysis, I would recommend shortening it.

Reviewer #2 (Comments for the Author):

Földi et al. submitted a well written manuscript on a novel gene acting in fruiting body development of the model fungus *Coprinopsis cinerea*. The paper is nicely to read, the experiments on the whole very well described and convincing, and the discussion is interesting setting former knowledge and newly received views by the current work nicely into relation. I fully support the publication of the manuscript. There are also some minor comments given below for the attention of the authors to improve or correct here and there some smaller points. For the benefit of the reader who is potentially not familiar with the genetic *Coprinopsis* system, a few words should be added on strain AmutBmut and terms related to the developmental pathway of fruiting body formation may need short explanations.

Specific comments:

- I wonder whether snb (or rather also here snb1) as first word in the sentence starting in line 24 and in line 40, also in lines 514, 548 and 594 should not also be written in lower case, because it is a given gene name
- 94: PriA here in italic as the other genes mentioned in the paragraph?
- 95-96: mentioning that this was found in *S. commune*?
- 102: is it meaningful to call such very early structure in development 'fruiting body'? I found this too often to confuse people (students)
- 116: is it needed to explain the word hyphal knots and primordia?
- stage, 1, 2, 3 primordia?
- 122: pab1 is the selection marker because of a defect in the transformed strain that was not a normal monokaryon? Would be better to add here a few words on strain used and its fruiting abilities ...
- 134: abbreviation wt needs definition?
- 150: that were developed?
- 231, 259, etc: letter p in italic?
- 310: GH20 stands specifically for ...?
- 458: of its thioredoxin gene
- 462: D-arabinono-1,4-lactone
- 465: S in glutathione-S-transferase in italic
- 479: downregulated (ID:
- 485: letter O in enzyme name written here in italic should then not be written in italic
- 486: but O here in italic
- 492: are there low expressed anyhow?
- 512: fruiting bodies of Agaricomycetes. (shown to be not found in Ascomycetes - thus, their fruiting body formation must be independent of snb1)
- 535: but not in fruiting-body forming ascomycetes specify clearly
- 537: Such annotated ...
- 539: highly variable in sequence
- 560, sentence structure unclear: domains have a clear
- 636: p for para in italic
- 823, 884: volume number?
- 887: full information?
- 1010: full boom information? Publisher? Is it a Mycota book?
- 1035, meaning of: 7. ?
- 1042: correct writing?
- to understand Fig. 2: in 2a, the base (bottom) of the structures are at the left of the photos - is this also the case in 2b, or are the structures turned by 90{degree sign}C around (basically with bottom at the bottom of the photos?) relative to the structures in 2a. Is the brown bit in the lower part the place corresponding to the normal cap? Are a and b photographed in same size relationship (size if bar in 2b? in 2a it should be the same size gar?)
- Fig. 3: have the plates seen light during mycelial growth (e.g. by shortly daily crosschecking)? The different ring structures on the mycelium may suggest so? Please add missing information on illumination parameters into the legend
- Seeing the structures in Fig. 3 - I wonder that some basic information in location of stipe and cap still remains in the mutant? Outer veil development still happens? Inner veil development connective between cap and stipe (by lack of cap development)

probably not?

- Fig. 4b: Inside, Outside labels could improve the figure
- Fig. 4c: In the legend, *snb1* as gene should be written in italic
- Fig. 5: please crosscheck in that the first letter of the first word is always a capital (e.g. currently is in lower case: Chromatin-related genes)
- 1193: et al. fully in italic (or fully not?)
- 1195: letters p in italic?
- legends Fig. S1: gene name *snb1* should here not be in italic in a text written in italic ... (or change vice versa the legend text into normal letters but not the gene name ...); what means I., II. and III. above the lanes? The three different transformants obtained?
- Fig. S2: scale bar - size in legend is different from sizes in photos; there are no label as a, b, c and d in the figure
- legend Fig. S3: polish the language please (e.g.: into the *snb1* gene ...)
- Fig. S4 legend - gene name in italic or is here the protein SNB1 meant? *C. cinerea* in italic
- Fig. S5, legend: letter p for p value in italic?
- Fig. S7, legend: *snb1* in italic

Reviewer #3 (Comments for the Author):

This manuscript describes phenotype and RNA-seq analysis of the newly isolated gene knock-out mutants of *snb1*, named after the mutant's morphological character like a snowball, in *Coprinopsis cinereus*. And it demonstrated that the gene product may have an important role in the fruiting body development in this fungus as well as most provably in other mushroom-forming fungi. This work highlights a successful utilization of a reverse genetic analysis for a gene encoding an unannotated protein found in the homology search analysis, indicting its actual requirement in fruiting body development.

The experiments were well-designed and done carefully. The disruption of the gene was checked by amplification of a specific fragment in genomic PCR experiment as well as complement analysis by introducing wt *snb1* gene to the mutants, that ensured that the phenotype they observed are result of the disruption of the gene not from randomly happened chromosome rearrangements or off-target effects of the CRISPR/Cas9. Furthermore, the results of the RNA-seq were described in a 'sober' and accurate way. Some time, postgenomics tempts people to assert a causal relationship between the phenotype and differences in gene expression. But this manuscript describes just observed results indifferently and there seems no major defects to assert such an arbitrary speculation, which is very important. Function of a gene nominated from postgenomic analysis should be tested experimentally, for example through a reverse genetic approach as described in this manuscript. There seems to be no major points to be addressed.

Minor comments would be:

l.129 and many other places, the Greek character 'delta' should not be in Italics, because it is not a part of a gene nomenclature which must be typed in Italics. Please unify the way to type the '*delta snb1*', like the ones in line 258 or 358.

l.337, '*snb1*' should be in Italics.

l.378, 'suggest that these genes play a role in the development of basidiomycetes' seems to assert bit too much, cause no direct evidence exists to say so. It would be better to bevel the expression.

l.500, what does 'has a central role in fruiting body development'? Maybe SNB1, right? I do not think that 'the disruption of *snb1*' has such a role. Please specify and rephrase the subject for this phrase.

Finally, how do you think about DEGs between wt and the mutant strains in vegetative growth? As you wrote, the expression of *snb1* were negligible in the vegetative phase and there were no changes in the growth rate, is it plausible to expect there are no DEGs in vegetative growth phase? This can be commented either in results or discussion.

Földi et al. submitted a well written manuscript on a novel gene acting in fruiting body development of the model fungus *Coprinopsis cinerea*. The paper is nicely to read, the experiments on the whole very well described and convincing, and the discussion is interesting setting former knowledge and newly received views by the current work nicely into relation. I fully support the publication of the manuscript. There are also some minor comments given below for the attention of the authors to improve or correct here and there some smaller points. For the benefit of the reader who is potentially not familiar with the genetic *Coprinopsis* system, a few words should be added on strain AmutBmut and terms related to the developmental pathway of fruiting body formation may need short explanations.

Specific comments:

- I wonder whether *snb* (or rather also here *snb1*) as first word in the sentence starting in line 24 and in line 40, also in lines 514, 548 and 594 should not also be written in lower case, because it is a given gene name
- 94: *PriA* here in italic as the other genes mentioned in the paragraph?
- 95-96: mentioning that this was found in *S. commune*?
- 102: is it meaningful to call such very early structure in development 'fruiting body'? I found this too often to confuse people (students)
- 116: is it needed to explain the word hyphal knots and primordia?
- stage, 1, 2, 3 primordia?
- 122: *pab1* is the selection marker because of a defect in the transformed strain that was not a normal monokaryon? Would be better to add here a few words on strain used and its fruiting abilities ...
- 134: abbreviation *wt* needs definition?
- 150: that were developed?
- 231, 259, etc: letter *p* in italic?
- 310: GH20 stands specifically for ...?
- 458: of its thioredoxin gene
- 462: D-arabinono-1,4-lactone
- 465: *S* in glutathione-S-transferase in italic
- 479: downregulated (ID:
- 485: letter *O* in enzyme name written here in italic should then not be written in italic
- 486: but *O* here in italic
- 492: are there low expressed anyhow?
- 512: fruiting bodies of Agaricomycetes. (shown to be not found in Ascomycetes – thus, their fruiting body formation must be independent of *snb1*)
- 535: but not in fruiting-body forming ascomycetes specify clearly

- 537: Such annotated ...
- 539: highly variable in sequence
- 560, sentence structure unclear: domains have a clear
- 636: p for para in italic
- 823, 884: volume number?
- 887: full information?
- 1010: full boom information? Publisher? Is it a Mycota book?
- 1035, meaning of: 7. ?
- 1042: correct writing?
- to understand Fig. 2: in 2a, the base (bottom) of the structures are at the left of the photos – is this also the case in 2b, or are the structures turned by 90° around (basically with bottom at the bottom of the photos?) relative to the structures in 2a. Is the brown bit in the lower part the place corresponding to the normal cap? Are a and b photographed in same size relationship (size if bar in 2b? in 2a it should be the same size gar?)
- Fig. 3: have the plates seen light during mycelial growth (e.g. by shortly daily crosschecking)? The different ring structures on the mycelium may suggest so? Please add missing information on illumination parameters into the legend
- Seeing the structures in Fig. 3 – I wonder that some basic information in location of stipe and cap still remains in the mutant? Outer veil development still happens? Inner veil development connective between cap and stipe (by lack of cap development) probably not?
- Fig. 4b: Inside, Outside labels could improve the figure
- Fig. 4c: In the legend, *snb1* as gene should be written in italic
- Fig. 5: please crosscheck in that the first letter of the first word is always a capital (e.g. currently is in lower case: Chromatin-related genes)
- 1193: et al. fully in italic (or fully not?)
- 1195: letters p in italic?
- legends Fig. S1: gene name *snb1* should here not be in italic in a text written in italic ... (or change vice versa the legend text into normal letters but not the gene name ...); what means I., II. and III. above the lanes? The three different transformants obtained?
- Fig. S2: scale bar - size in legend is different from sizes in photos; there are no label as a, b, c and d in the figure
- legend Fig. S3: polish the language please (e.g.: into the *snb1* gene ...)
- Fig. S4 legend – gene name in italic or is here the protein SNB1 meant? *C. cinerea* in italic
- Fig. S5, legend: letter p for p value in italic?
- Fig. S7, legend: *snb1* in italic

Reviewer #1 (Comments for the Author):

Földi and co-authors have identified the gene snowball (*snb1*) as an important gene in the development of the mushroom-forming fungus *Coprinopsis cinerea*. It is considered unannotated and was primarily selected based on its expression profile in previously published data sets, as well as strong conservation in other mushroom-forming fungi. A gene deletion of *snb1* resulted in stunted development. Next, RNA-Seq was performed to identify genes downstream of *snb1*. In general, the research appears to have been performed correctly. I have a few minor comments.

The authors consider the gene/protein snowball as 'unannotated', even though they report that it has 7 transmembrane domains and even a PFAM domain. Although the PFAM domain is essentially a domain of unknown function, I still don't consider this protein to be completely unannotated. A better term could be a putative transmembrane protein of unknown function, or something like that.

ANSWER: We appreciate the Reviewer's valuable comments, which led us to reconsider our usage of the term 'unannotated'. While we cannot predict or deduce the function of the snb1 gene, labeling it as an 'unannotated gene' was indeed inaccurate, because it possesses a conserved domain, specifically the DUF6533, without a known function. We have taken the reviewer's advice and made the necessary corrections to the manuscript.

In my opinion the RNA-Seq dataset was not very useful to answer questions about the function of snowball. The authors carefully avoid stating that the transcriptomic changes are directly due to the deletion of snowball, which is good. Instead, they take the altered morphology as an opportunity to study genes involved in development. In principle that is fine, but in retrospect it would have been better to do RNA-Seq on a timepoint when WT and delta-*snb1* first diverged in development (or even a bit before), in order to capture the early responses to the deletion.

ANSWER: We chose the secondary hyphal knots as the first sample time point, although the Δsnb1 secondary hyphal knots exhibited larger size than those of the wild type, visible tissue differentiation had not yet initialized in any strains in that stage. Furthermore, based on previous transcriptomic studies, the expression level of snb1 was negligible in the vegetative mycelia, therefore hyphal knots might be even more suitable, but their manual sampling without risking significant VM contamination proved technically infeasible. We indicated this information in the manuscript.

Over half the Results is dedicated to this RNA-Seq analysis, I would recommend shortening it.

ANSWER: The RNA-Seq analysis was conducted not to infer the function of snb1, but to understand the expression patterns associated with the differentiation of fruiting bodies, which we think the section achieves. Therefore we consider the RNA-Seq results as the central part of our manuscript. We shortened this part slightly while maintaining its significance.

Reviewer #2 (Comments for the Author):

Földi et al. submitted a well written manuscript on a novel gene acting in fruiting body development of the model fungus *Coprinopsis cinerea*. The paper is nicely to read, the experiments on the whole very well described and convincing, and the discussion is interesting setting former knowledge and newly received views by the current work nicely into relation. I

fully support the publication of the manuscript. There are also some minor comments given below for the attention of the authors to improve or correct here and there some smaller points. For the benefit of the reader who is potentially not familiar with the genetic *Coprinopsis* system, a few words should be added on strain AmutBmut and terms related to the developmental pathway of fruiting body formation may need short explanations.

ANSWER: *We appreciate the Reviewer's evaluation of our manuscript, which helped to improve the clarity and quality. We corrected all the mistakes according to their recommendations.*

Specific comments:

- I wonder whether snb (or rather also here snb1) as first word in the sentence starting in line 24 and in line 40, also in lines 514, 548 and 594 should not also be written in lower case, because it is a given gene name

ANSWER: *We corrected the sentence.*

- 94: PriA here in italic as the other genes mentioned in the paragraph?

ANSWER: *We corrected it.*

- 95-96: mentioning that this was found in *S. commune*?

ANSWER: *We made the correction.*

- 102: is it meaningful to call such very early structure in development 'fruiting body'? I found this too often to confuse people (students)

ANSWER: *This sentence refers to our statement about snb1 may play a role in fruiting body development of Agaricomycetes based on the developmental expression pattern comparison of different species. We suspect the Reviewer refers to secondary hyphal knots and stage 1 primordia in this comment. We agree that these are early stages of development, nevertheless, in this sentence we refer not only to early stages, but also to later ones, so we think the term fruiting body is warranted here.*

- 116: is it needed to explain the word hyphal knots and primordia?

ANSWER: *We defined in detail these stages in the Results chapter.*

- stage, 1, 2, 3 primordia?

ANSWER: *We clarified the stages: "In *C. cinerea* the expression of snb1 is negligible in vegetative mycelia, but increased 144- and 240-fold in secondary hyphal knots and stage 1 primordia, respectively (26)."*

- 122: *pab1* is the selection marker because of a defect in the transformed strain that was not a normal monokaryon? Would be better to add here a few words on strain used and its fruiting abilities ...

ANSWER: *We included this information.*

- 134: abbreviation wt needs definition?

ANSWER: *We defined wt.*

- 150: that were developed?

ANSWER: *We deleted it.*

- 231, 259, etc: letter p in italic?

ANSWER: *We corrected them.*

- 310: GH20 stands specifically for ...?

ANSWER: *We explained it.*

- 458: of its thioredoxin gene

ANSWER: *We corrected it.*

- 462: D-arabinono-1,4-lactone

ANSWER: *We corrected it.*

- 465: S in glutathione-S-transferase in italic

ANSWER: *We corrected it.*

- 479: downregulated (ID:

ANSWER: *We deleted it.*

- 485: letter O in enzyme name written here in italic should then not be written in italic

ANSWER: *We corrected it.*

- 486: but O here in italic

ANSWER: *We corrected it.*

- 492: are there low expressed anyhow?

ANSWER: *We deleted the related sentence because it was not relevant.*

- 512: fruiting bodies of Agaricomycetes. (shown to be not found in Ascomycetes - thus, their fruiting body formation must be independent of snb1)

ANSWER: *We corrected it.*

- 535: but not in fruiting-body forming ascomycetes specify clearly

ANSWER: *We corrected it.*

- 537: Such annotated ...

ANSWER: *We deleted it.*

- 539: highly variable in sequence

ANSWER: *We corrected it.*

- 560, sentence structure unclear: domains have a clear

ANSWER: *We corrected it.*

- 636: p for para in italic

ANSWER: *We corrected it.*

- 823, 884: volume number?

ANSWER: *We corrected them.*

- 887: full information?

ANSWER: *We corrected it.*

- 1010: full boom information? Publisher? Is it a Mycota book?

ANSWER: *We specified it.*

- 1035, meaning of: 7. ?

ANSWER: *We deleted it.*

- 1042: correct writing?

ANSWER: *We corrected it.*

- to understand Fig. 2: in 2a, the base (bottom) of the structures are at the left of the photos - is this also the case in 2b, or are the structures turned by 90{degree sign}C around (basically with bottom at the bottom of the photos?) relative to the structures in 2a. Is the brown bit in the lower part the place corresponding to the normal cap? Are a and b photographed in same size relationship (size if bar in 2b? in 2a it should be the same size gar?)

ANSWER: *We appreciate the reviewer's valuable comment. Yes, the wt dark stipes are turned by 90°, we included a note on this in the legend. We accidentally omitted the legend of the size bar (size bar: 2mm). a) and b) are photographed in the same size, therefore the size bar is applicable for both. We corrected the legend.*

- Fig. 3: have the plates seen light during mycelial growth (e.g. by shortly daily crosschecking)? The different ring structures on the mycelium may suggest so? Please add missing information on illumination parameters into the legend

ANSWER: *Yes, in this case the vegetative mycelia of the strains were grown under alternating dark and light cycles, leading to the formation of ring-like structures on the mycelia. We included this information in the legend.*

- Seeing the structures in Fig. 3 - I wonder that some basic information in location of stipe and cap still remains in the mutant? Outer veil development still happens? Inner veil development connective between cap and stipe (by lack of cap development) probably not?

ANSWER: *In the manuscript we added our observations regarding this comment. We used the term 'universal veil' for 'outer veil' and 'partial veil' for 'inner veil'. Based on the section photos, we concluded that the mutant is unable to form stipe or cap structures, even in the presence of LiCl. It appeared that the universal veil developed, but due to the absence of cap, the partial veil was missing. We added this information in the Results chapter. A region of higher cell density developed in the mutant where one would expect the cap tissues, however, we feel calling these even cap rudiments would be potentially ungrounded. snb1 mutant strains growing*

in agar plates containing 0.5-1.5g/l LiCl produce fruiting bodies with elongated basal nodules. We included this information in the Results.

- Fig. 4b: Inside, Outside labels could improve the figure

ANSWER: *We added these labels to the figure.*

- Fig. 4c: In the legend, *snb1* as gene should be written in italic

ANSWER: *We corrected it.*

- Fig. 5: please crosscheck in that the first letter of the first word is always a capital (e.g. currently is in lower case: Chromatin-related genes)

ANSWER: *We corrected it.*

- 1193: et al. fully in italic (or fully not?)

ANSWER: *We fully italicized it.*

- 1195: letters p in italic?

ANSWER: *We corrected it.*

- legends Fig. S1: gene name *snb1* should here not be in italic in a text written in italic ... (or change vice versa the legend text into normal letters but not the gene name ...); what means I., II. and III. above the lanes? The three different transformants obtained?

ANSWER: *We corrected the gene name. Lanes I., II. and III. mean the independent $\Delta snb1$ strains, we indicated this information in the figure legend.*

- Fig. S2: scale bar - size in legend is different from sizes in photos; there are no label as a, b, c and d in the figure

ANSWER: *We corrected the labels and the legend regarding scale bars.*

- legend Fig. S3: polish the language please (e.g.: into the *snb1* gene ...)

ANSWER: *We corrected it.*

- Fig. S4 legend - gene name in italic or is here the protein SNB1 meant? *C. cinerea* in italic

ANSWER: *We corrected the legend and the title of the supplementary figure to “Phylogenetic tree of the SNB1 family”. Therefore, we used SNB1 instead of snb1.*

- Fig. S5, legend: letter p for p value in italic?

ANSWER: *We corrected it.*

- Fig. S7, legend: *snb1* in italic

ANSWER: *We corrected it.*

Reviewer #3 (Comments for the Author):

This manuscript describes phenotype and RNA-seq analysis of the newly isolated gene knock-out mutants of *snb1*, named after the mutant's morphological character like a snowball, in

Coprinopsis cinereus. And it demonstrated that the gene product may have an important role in the fruiting body development in this fungus as well as most provably in other mushroom-forming fungi. This work highlights a successful utilization of a reverse genetic analysis for a gene encoding an unannotated protein found in the homology search analysis, indicating its actual requirement in fruiting body development.

The experiments were well-designed and done carefully. The disruption of the gene was checked by amplification of a specific fragment in genomic PCR experiment as well as complement analysis by introducing wt *snb1* gene to the mutants, that ensured that the phenotype they observed are result of the disruption of the gene not from randomly happened chromosome rearrangements or off-target effects of the CRISPR/Cas9. Furthermore, the results of the RNA-seq were described in a 'sober' and accurate way. Some time, postgenomics tempts people to assert a causal relationship between the phenotype and differences in gene expression. But this manuscript describes just observed results indifferently and there seems no major defects to assert such an arbitrary speculation, which is very important. Function of a gene nominated from postgenomic analysis should be tested experimentally, for example through a reverse genetic approach as described in this manuscript. There seems to be no major points to be addressed. **ANSWER:** *We appreciate the Reviewer's valuable suggestions, which helped to improve the clarity and quality. We corrected all the mistakes according to the recommendations.*

Minor comments would be:

1.129 and many other places, the Greek character 'delta' should not be in Italics, because it is not a part of a gene nomenclature which must be typed in Italics. Please unify the way to type the 'delta *snb1*', like the ones in line 258 or 358.

ANSWER: *We corrected it in the whole text.*

1.337, '*snb1*' should be in Italics.

ANSWER: *We corrected it.*

1.378, 'suggest that these genes play a role in the development of basidiomycetes' seems to assert bit too much, cause no direct evidence exists to say so. It would be better to bevel the expression.

ANSWER: *We deleted this part, because as the reviewer mentioned, there is no direct evidence to assume it.*

1.500, what does 'has a central role in fruiting body development'? Maybe *SNB1*, right? I do not think that 'the disruption of *snb1*' has such a role. Please specify and rephrase the subject for this phrase.

ANSWER: *According to the reviewer's comment, we rephrased this sentence: "Taken together, the RNA-Seq data presented here for two developmental stages of wt and $\Delta snb1$ strains suggest that the disruption of *snb1* affects numerous molecular pathways and ***SNB1* might** have a central role in fruiting body development."*

Finally, how do you think about DEGs between wt and the mutant strains in vegetative growth? As you wrote, the expression of *snb1* were negligible in the vegetative phase and there were no

changes in the growth rate, is it plausible to expect there are no DEGs in vegetative growth phase? This can be commented either in results or discussion.

ANSWER: *Yes, we expect to not find DEGs in the vegetative phase, this is why we did not perform this analysis. We added this information to the manuscript.*

Re: mSystems01208-23R1 (Snowball: a novel gene family required for developmental patterning of fruiting bodies of mushroom-forming fungi (Agaricomycetes))

Dear Dr. László Nagy:

I am pleased to inform you that your manuscript has been accepted for publication in mSystems.

Your manuscript has been forwarded to the ASM production staff for publication. Your paper will first be checked to make sure all elements meet the technical requirements. ASM staff will contact you if anything needs to be revised before copyediting and production can begin. Otherwise, you will be notified when your proofs are ready to be viewed.

Featured Image Submissions: If you would like to submit a potential Featured Image, please email a file and a short legend to mSystems@asmusa.org. Please note that we can only consider images that (i) the authors created or own and (ii) have not been previously published. By submitting, you agree that the image can be used under the same terms as the published article. File requirements: square dimensions (4" x 4"), 300 dpi resolution, RGB colorspace, TIF file format.

Sincerely,
Benjamin Wolfe
Senior Editor
mSystems